# Cannistraci-Hebb Training on Ultra-Sparse Spiking Neural Networks

**Yuan Hua[1,†], Jilin Zhang[1,†], Yingtao Zhang[2,3], Leyi You[4], Baobo Xiong[5], Carlo Vittorio Cannistraci[2,3,6,\*], Hong Chen[1,\*]**

[1] School of Integrated Circuits, Tsinghua University, Beijing China
[2] Center for Complex Network Intelligence at the Tsinghua Laboratory of Brain and Intelligence, Department of Psychological and Cognitive Sciences
[3] Department of Computer Science and Technology, Tsinghua University, Beijing, China
[4] Department of Physics, Tsinghua University, Beijing, China
[5] Zhili College, Tsinghua University, Beijing, China
[6] School of Biomedical Engineering, Tsinghua University, Beijing, China
[\*] Corresponding authors
[†] Equal contribution

`kalokagathos.agon@gmail.com, hongchen@tsinghua.edu.cn`

## Abstract

Inspired by the brain's spike-based computation, spiking neural networks (SNNs) inherently possess temporal activation sparsity. However, when it comes to the sparse training of SNNs in the structural connection domain, existing methods fail to achieve ultra-sparse network structures without significant performance loss, thereby hindering progress in energy-efficient neuromorphic computing. This limitation presents a critical challenge: how to achieve high levels of structural connection sparsity while maintaining performance comparable to fully connected networks. To address this challenge, we propose the Cannistraci-Hebb Spiking Neural Network (CH-SNN), a novel and generalizable dynamic sparse training framework for SNNs consisting of four stages. First, we propose a sparse spike correlated topological initialization (SSCTI) method to initialize a sparse network based on node correlations. Second, temporal activation sparsity and structural connection sparsity are integrated via a proposed sparse spike weight initialization (SSWI) method. Third, a hybrid link removal score (LRS) is applied to prune redundant weights and inactive neurons, improving information flow. Finally, the CH3-L3 network automaton framework inspired by Cannistraci-Hebb learning theory is incorporated to perform link prediction for potential synaptic regrowth. These mechanisms enable CH-SNN to achieve sparsification across all linear layers. We have conducted extensive experiments on six datasets including CIFAR-10 and CIFAR-100, evaluating various network architectures such as spiking convolutional neural networks and Spikformer. The proposed method achieves a maximum sparsity of 97.75% and outperforms the fully connected (FC) network by 0.16% in accuracy. Furthermore, we apply CH-SNN within an SNN training algorithm deployed on an edge neuromorphic processor. The experimental results demonstrate that, compared to the FC baseline without CH-SNN, the sparse CH-SNN architecture achieves up to 98.84% sparsity, an accuracy improvement of 2.27%, and a 97.5× reduction in synaptic operations, and the energy consumption is reduced by an average of 55× across four datasets. Our code is available at https://github.com/HuaGuaiGuai/CH-SNN.

## 1 Introduction

The increasing computational demands and energy consumption of deep neural networks have spurred the exploration of energy-efficient alternatives. Inspired by the event-driven processing mechanism of the human brain, spiking neural networks (SNNs) have emerged as a promising solution due to their inherent temporal activation sparsity (Li et al., 2024; Roy et al., 2019). The temporal

activation sparsity of SNNs stems from their spiking characteristics—neurons only fire spikes when the membrane potential reaches the threshold, remaining in a resting state for most of the time (Wu et al., 2023). Compared to artificial neural networks (ANNs), SNNs demonstrate significant advantages in energy efficiency, making them well-suited for a variety of edge-side applications such as gas detection (Huo et al., 2025), sEMG-based gesture recognition (Zhang et al., 2024a) and real-time multi-object recognition (Merolla et al., 2014).

Despite the inherent advantages in temporal activation sparsity, SNNs often suffer from a fixed architecture that lacks flexibility and structural plasticity, which limits the learning capability of SNNs and their application in resource-limited neuromorphic hardware (Davies et al., 2018). To address this problem, previous works introduce structural connection sparsity through network pruning and regrowth (Bennett et al., 2018). Although sparse training has proven effective in reducing parameter counts and improving computational efficiency in ANNs (Mocanu et al., 2018; Evci et al., 2019; Yuan et al., 2021; Zhang et al., 2022), its application in SNNs remains challenging. This is due to the spike-based computation and the non-differentiable nature of the spiking activation function of SNNs, which hinder direct gradient-based optimization. Consequently, most sparse training methods for ANNs that rely on gradient information cannot be directly applied to SNNs.

Specifically, current research in sparse SNNs training methods faces a significant challenge in achieving high levels of structural connection sparsity while maintaining performance comparable to that of their fully connected counterpart. For instance, the adaptive structural development of SNN (SD-SNN) model (Han et al., 2025a), incorporates multiple brain-inspired developmental mechanisms, including synaptic elimination, neuronal pruning and synaptic regeneration. Besides, it also uses adaptive pruning and regrowth rates, which led to the structural stability. As a result, SD-SNN achieves 98.56% accuracy with 1.45% improvement on DVS-Gesture dataset, but only reaches a maximum sparsity of 61.10%. Similarly, Shen et al. (2025) propose a two-stage dynamic structure learning method, effectively addressing the limitations of fixed pruning ratios and static sparse training methods prevalent in existing models. Nonetheless, their approach attains an average structural connection sparsity of around 70%. Some studies directly apply ANN-based sparse training methods to SNNs, Gradient Rewiring (Shen et al., 2025) method achieves up to 90% sparsity. However, it exhibits an accuracy degradation of 3.55% compared to its fully connected counterpart.

To address this challenge, we propose the C̲annistraci-H̲ebb S̲piking N̲eural N̲etwork (CH-SNN), a novel and generalizable dynamic sparse training framework for SNNs, which achieves high levels of structural connection sparsity and maintaining performance comparable to that of its fully connected (FC) counterpart. The main contributions of this work are summarized as follows:

- **Introducing a novel sparse training framework.** We propose a four-stage dynamic sparse training framework (CH-SNN) consisting of sparse topology initialization, sparse weight initialization, network pruning and network regrowth. CH-SNN attains 99% structural connection sparsity in all linear layers and shows better performance than FC networks on the CIFAR-100, MNIST, N-MNIST, CIFAR10-DVS and DVS-Gesture datasets respectively.

- **Proposing efficient initialization methods.** We propose two initialization methods, one is Sparse Spike Correlated Topological Initialization (SSCTI) which initializes an ultra-sparse network structure by leveraging correlations among input nodes, another is Sparse Spike Weight Initialization (SSWI) which incorporates temporal activation sparsity and structural connection sparsity of SNNs to initialize weights. SSCTI and SSWI enhance the performance of the link predictor and facilitate faster training from the initial phases.

- **Demonstrating superior performance across architectures and datasets.** We have conducted extensive experiments, the experimental results demonstrate that CH-SNN outperforms existing sparse SNN training methods across six datasets (CIFAR10-DVS, CIFAR-10, CIFAR-100, MNIST, N-MNIST and DVS-Gesture) and three network structures. Notably, it attains a 0.16% accuracy improvement over the FC network at a sparsity of 97.75%. We apply CH-SNN to a hardware-friendly algorithm S-TP, which has been implemented on a neuromorphic processor for edge-side AI applications. Experimental results show that CH-SNN significantly improve energy efficiency, achieving an average improvement of $55\times$ across four datasets.

## 2 RELATED WORKS

### 2.1 SPARSE SPIKING NEURAL NETWORKS

Structural connection sparsification is one of the key technologies for enhancing SNNs energy efficiency. By reducing redundant links and neurons within the model, it can significantly reduce computational and storage overhead. Existing sparse SNNs training methods can be categorized into pruning and sparse training.

**Pruning.** Pruning methods initialize a fully connected network structure and gradually remove insignificant links during training. Current SNNs pruning approaches can be divided into the two types: (1) Biological plasticity pruning. This approach draws inspiration from the developmental mechanisms of the brain, leveraging biological synaptic plasticity to accomplish the pruning of SNNs. Han et al. (2025b;a) propose the developmental plasticity-inspired adaptive pruning method, which takes into account multiple biologically realistic mechanisms, so that the network structure can be dynamically optimized. Rathi et al. (2019) present a sparse SNN training method where pruning is based on the spike timing dependent plasticity model (STDP). Links between pre-neuron and post-neuron with low correlation or uncorrelated spiking activity are pruned. Liu et al. (2022) propose a dynamic pruning framework (named DynSNN) for SNNs, enabling dynamic optimization of the network topology. (2) Transfer ANNs pruning method to SNNs. These methods adapt pruning techniques from ANNs. For instance, Chen et al. (2022) use different functions describing the growing threshold of state transition to regulate the pruning speed, avoiding disastrous performance degradation at the final stage of training. Deng et al. (2023) formulate the link pruning problem as a constrained optimization problem, which is addressed by integrating spatiotemporal backpropagation (STBP) with the alternating direction method of multipliers (ADMMs). Backpropagation with sparsity regularization (BPSR) (Yan et al., 2022) incorporates an $L_1$ regularization term into the loss function to drive the weights toward zero, followed by a static threshold-based pruning method, thereby achieving network structural connection sparsification.

**Sparse training.** In contrast to pruning methods, sparse training begins with a sparsely connected network and dynamically alternates between pruning less important connections and growing new ones during learning. It maintains sparsity in both the forward and backward propagation during the training process, resulting in lower hardware requirements. For example, the Deep Rewiring (Deep R) (Bellec et al., 2018) method prunes links when their value changes sign during updates, and randomly regenerate an equivalent number of links. This process is repeated over multiple rounds. Based on Deep R, Chen et al. (2021) introduce a Gradient Rewiring (Grad R) approach to further modify the gradient values of the links, enabling previously pruned links to regenerate. Furthermore, Shen et al. (2025) propose a two-stage dynamic structure learning method for deep SNNs, the first stage evaluates the network's compressibility based on the PQ index (Diao et al., 2023) and adaptively determines the regrowth ratio, and the second stage performs pruning and regrowth according to this ratio. Qi et al. (2018) propose a spiking neural network with connection gates (SNN-CG) to jointly learn the topology and the weights in SNN. The connection structures and the weights are learned alternately until a termination condition is satisfied. Neurogenesis dynamics-inspired spiking neural network (NDSNN) training method (Huang et al., 2023) trains a model from scratch using dynamic sparsity. NDSNN creates a drop-and-grow strategy to promote link reduction. Based on RigL (Evci et al., 2019), Lasby et al. (2024) propose a sparse-to-sparse dynamic sparse training method named Structured RigL (SRigL), which learns a sparse neural network with constant fan-in fine-grained structured sparsity while maintaining generalization comparable with RigL.

### 2.2 CANNISTRACI-HEBB THEORY AND NETWORK TOPOLOGY INTELLIGENCE

Inspired by the dynamic sparse connectivity characteristics of the brain, the Cannistraci-Hebb (CH) theory (Cannistraci et al., 2013; Daminelli et al., 2015; Cannistraci, 2018a; Muscoloni et al., 2018; Zhao et al., 2025) is a general theoretical framework developed in the field of network science to predict the non-observed dynamic connectivity of complex networks, using the mere knowledge of the network topology. CH theory is also recently introduced (Zhao et al., 2025; Zhang et al., 2024b) for dynamic sparse training for deep AI, demonstrating a gradient-free link regrowth mechanism that relies solely on topological information. For example, Cannistraci-Hebb Training (CHT) (Zhang et al., 2024d) is applied to ANNs, utilizing the CH3-L3 network automaton for link prediction. CH3-L3 is one of the highest-performing and most robust network automata under the Cannistraci-Hebb the-

ory (Zhao et al., 2025). It can automatically evolve the network topology of a given structure by identifying node pairs with the fewest external connections within the local community structure, thereby guiding link regrowth. For multiple tasks, CHT achieves better performance surpassing than fully connected networks with only 1% connections, demonstrating the ultra-sparse advantage. Importantly, CHT is shown to induce during training also a node sparsification process (called network node percolation), which at the end of the training compressed the node size of certain networks to around the 30 percent of the initial size, preserving or improving task performance. Furthermore, Zhang et al. (2024c) put forward a Cannistraci-Hebb Training soft rule (CHTs) which probabilistically removes network links based on a removal fraction and regrows new links according to CH3-L3 prediction scores, overcoming CHT's tendency to fall into epitopological local minima during the early stages of training when topological noise is significant.

## 3 METHODS

### 3.1 SPIKING NEURAL NETWORK

**Fundamentals.** Unlike artificial neural networks, SNNs use sparse spike signals to transmit information. The spike signals enable SNNs to avoid Multiply-Accumulate (MAC) operations. Thereby reducing energy consumption and computational load (Rueckauer et al., 2017). In this paper, we adopt the leaky integrate-and-fire (LIF) neuron (Abbott, 1999) to process spike signals, as shown in Equation (1).

$$v_j(t+1) = (1 - z_j(t))\alpha v_j(t) + \sum_i W_{ij} x_i(t+1), \quad z_j(t) = U(v_j(t) - \theta) \tag{1}$$

where $t$ denotes the time step, $v_j(\cdot)$ represents the membrane potential of the neuron $j$, $\alpha$ is the membrane potential decay constant, $W_{ij}$ denotes the synaptic weight, $x_i(t)$ is the input spike, and $U$ is the step function. When the membrane potential accumulates and exceeds the firing threshold $\theta$, the neuron emits an output spike, denoted by $z_j(\cdot) = 1$. Otherwise, the neuron remains silent, i.e., $z_j(\cdot) = 0$. After firing a spike, the membrane potential is reset to zero.

**Training method.** It is challenging to apply standard gradient descent to SNNs. The step function in Equation (1) results in a derivative that is zero almost everywhere and undefined at the threshold point. This makes the direct calculation of partial derivatives $\frac{\partial z_j(\cdot)}{\partial v_j(\cdot)}$ impossible using conventional calculus, which prevents the application of the backpropagation algorithm. Thus we use the surrogate gradient method (Wu et al., 2018) to update the weights of SNNs.

**Sparse Target Propagation.** Sparse Target Propagation (S-TP) (Zhang et al., 2024a) adopts a hardware-friendly surrogate gradient method. S-TP randomly selects target windows in the learning process, reducing over 90% of the spike number in the learning process without noticeable accuracy degradation. S-TP has been implemented in a low-power neuromorphic processor, which proves its notable hardware-friendliness.

### 3.2 CANNISTRACI-HEBB SPIKING NEURAL NETWORK

In this paper, we propose the four-stage dynamic sparse training method for SNNs, named Cannistraci-Hebb Spiking Neural Network (CH-SNN). It is a general sparse training framework capable of sparsifying all linear layers in SNNs. The framework of CH-SNN is illustrated in Figure 1. **The first stage is sparse topology initialization.** We propose a sparse topology initialization method named Sparse Spike Correlated Topological Initialization (SSCTI) based on Pearson's phi coefficient to initialize an ultra-sparse network. **The second stage is sparse weight initialization.** We introduce the Sparse Spike Weight Initialization (SSWI) method, which incorporates the temporal activation sparsity, structural connection sparsity and neuronal threshold information of SNNs into the weight initialization process to perform weight initialization. **The third stage is network pruning.** We use a probability-based links pruning strategy to remove links according to a dynamic ratio $\zeta$. Subsequently, inactive neurons are identified and removed, according to the CHT network percolation procedure (Zhang et al., 2024d). **The fourth stage is network regrowth.** Here, the regrowth score of potential links is computed using CH3-L3. According to this score, links are regenerated with the same ratio applied in the network pruning stage, thereby maintaining the pre-

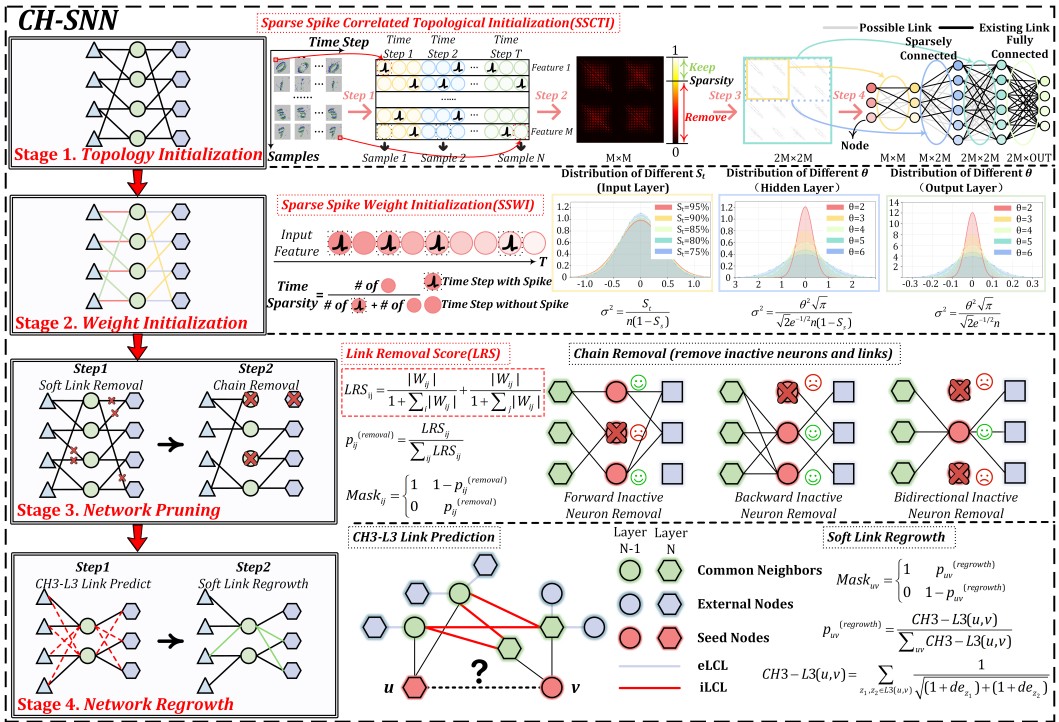

Figure 1: The framework of Cannistraci-Hebb Spiking Neural Network (CH-SNN).

defined structural connection sparsity. Links with higher regrowth scores are proportionally sampled according to the CHTs methodology (Zhang et al., 2024c).

### 3.2.1 SPARSE TOPOLOGY INITIALIZATION

As we know, the topology of a network should reflect the relationships between node features within some latent geometric space (Cannistraci & Muscoloni, 2020). The correlations between input features directly define the geometric relationships among nodes in this latent feature space. Therefore, by computing the correlations between nodes in the input layer, we preserve connections between highly correlated nodes according to the predefined sparsity. Thus, we propose the Sparse Spike Correlated Topological Initialization (SSCTI) method, as shown in Figure 1 Stage 1. Since the input of SNNs are discrete binary spike trains, we measure the correlation between input nodes using Pearson's phi coefficient (Pearson, 2015), which measures the strength and direction of association between two binary variables. We take each dimension of the input data $x_i$ and each time step $t$ as a variable and an independent sample, respectively. Thus, the total number of samples is $N \times T$. The Pearson's phi coefficient is described in Equation (2).

$$\phi_{ij} = \sqrt{\frac{\chi_{ij}^2}{2NT}} = \sqrt{\frac{\sum_{t=1}^{NT}(x_i(t)-E_i)^2/E_i + \sum_{t=1}^{NT}(x_j(t)-E_j)^2/E_j}{2NT}} \tag{2}$$

where $\phi_{ij}$ represents the Pearson's phi coefficient between input data $x_i$ and $x_j$, $M$ denotes the dimension of the input data, $T$ is total time step, $N$ stands for the number of samples, $\chi_{ij}^2$ represents the Chi-square statistic, and $E_i$ is the mean value of $x_i$. From Equation (2), we obtain the correlation matrix $\Phi \in \mathbb{R}^{M \times M}$, we keep the top $(1 - S_s)$ proportion of links of the SNN with the strongest correlations and remove other links in SNN, where $S_s$ stands for the structural connection sparsity, thereby completing the initialization of the network structure. The dimensionality of the hidden layer is determined by an expansion factor $\beta \geq 1$, $\beta \in \mathbb{Z}$, such that the dimension of the hidden layer equals the input dimension multiplied by $\beta$. This allows the hidden layer dimensionality to be flexibly adjusted, as shown in Figure 1 Stage 1. However, when CH-SNN is used to sparsify intermediate layers, such as linear layers within spiking convolutional neural networks (Lv et al., 2024)

or Spikformer (Zhou et al., 2022), the input distribution may be altered by preceding convolutional layers or attention layers. This makes it difficult for SSCTI to accurately capture feature correlations between nodes. To address this issue, we adopt a uniform random initialization strategy, which ensures that each node retains an equal number of connections. See more details in Appendix A.4.

### 3.2.2 SPARSE SPIKE WEIGHT INITIALIZATION

For ANNs, weight initialization strategies such as Kaiming initialization (He et al., 2015) are widely adopted. Most of these methods assume that weights follow a zero-mean Gaussian distribution, and determine the variance of this distribution under the principle of maintaining consistent variance of input data across layers. However, such approaches cannot be directly applied to the weight initialization in structural sparse networks. Although methods like SWI (Zhang et al., 2024d) have been proposed for sparse artificial neural networks, they are unsuitable for SNNs owing to their inability to incorporate temporal activation sparsity and the unique activation function of LIF neurons. To address this problem, we put forward the Sparse Spike Weight Initialization (SSWI) method, which incorporates the temporal activation sparsity ($S_t$), structural connection sparsity ($S_s$) and the neuronal threshold information ($\theta$) of SNNs into the weight initialization process. The detailed derivation is provided in Appendix A.1. The SSWI method is presented in Equation (3).

$$SSWI(W_{ij}^{(l)}) \sim \mathcal{N}(0, \sigma^2), \quad \sigma^2 = \begin{cases} \dfrac{S_t}{n(1-S_s)}, & (l=1) \\ \dfrac{\theta^2 \sqrt{\pi}}{\sqrt{2}e^{-1/2}n(1-S_s)}, & (1 < l < L) \\ \dfrac{\theta^2 \sqrt{\pi}}{\sqrt{2}e^{-1/2}n}, & (l=L) \end{cases} \tag{3}$$

where $S_t$ denotes the temporal activation sparsity of the input data in SNNs, $S_s$ represents the structural connection sparsity, $l$ is the index of the layer (with a total of $L$ layers), $n$ indicates the input feature dimension of the $l$ layer, and $\theta$ denotes the spike threshold in the LIF neuron. SSWI enhances training efficiency, leading to faster convergence from the initial phases.

### 3.2.3 NETWORK PRUNING

**Link Removal.** We propose a hybrid strategy to calculate the link removal score ($LRS$) that combines relative importance (RI) and weight magnitude (WM). The approach not only accelerates network sparsification but also promotes the activation of more neurons during training. The $LRS$ is defined as follows:

$$LRS_{ij}^{(l)} = \frac{|W_{ij}^{(l)}|}{1 + \sum_i |W_{ij}^{(l)}|} + \frac{|W_{ij}^{(l)}|}{1 + \sum_j |W_{ij}^{(l)}|} \tag{4}$$

where $LRS_{ij}^{(l)}$ denotes the link removal score of the weight $W_{ij}^{(l)}$, $\sum_j |W_{ij}^{(l)}|$ represents the sum of the magnitude of all weights connected to the input neuron $i$, and $\sum_i |W_{ij}^{(l)}|$ denotes the sum of the magnitude of all weights connected to the output neuron $j$. Instead of using the magnitude of the $LRS$ as the direct criterion for link removal, we sample from a multi-nomial distribution based on the $LRS$ value to determine whether a link should be removed. See more details in Appendix A.3.

**Chain Removal.** After link removal, neurons that are unilaterally or bilaterally disconnected (i.e., without any incoming or outgoing links) are regarded as inactive neurons. Since these inactive neurons lose the ability to transmit information, they may hinder information flow throughout the network. Because of the mechanism of CH3-L3, such inactive neurons are unable to regrow new links during the network regrowth stage. Therefore, during the chain removal step, we permanently remove them from the network. As illustrated in Figure 1 Stage 3, this process enhances overall information propagation.

### 3.2.4 NETWORK REGROWTH

We employ CH3-L3 to compute the link regrowth score for potential links, as CH3-L3 is recognized as the most robust and stable link predictor within the Cannistraci-Hebb theory (Zhang et al., 2024b;

Zhao et al., 2025; Zhang et al., 2025). To mitigate the risk of falling into epitopological local minima due to structural noise in the network, we sample from a binomial distribution based on the regrowth score to stochastically determine whether a link should be regenerated, instead of using the regrowth score directly as the criterion for link regrowth (Zhang et al., 2024c). The formula for calculating the link regrowth score is as follows:

$$\textbf{CH3-L3}(u, v) = \sum_{z_1, z_2 \in l3(u,v)} \frac{1}{\sqrt{(1 + de_{z_1}) \times (1 + de_{z_2})}} \tag{5}$$

where $u$ and $v$ are two nodes that may potentially form a link, and $z_1$, $z_2$ denote two intermediate nodes along a path of length 3 between $u$ and $v$—also referred to as common neighbor nodes of $u$ and $v$. The terms $de_{z_1}$ and $de_{z_2}$ represent the external local community connectivity degrees of nodes $z_1$ and $z_2$, respectively. A detailed description of the CH3-L3 is provided in Appendix A.2.

## 4 EXPERIMENTS

### 4.1 COMPREHENSIVE PERFORMANCE COMPARISON WITH OTHER METHODS

We compare our CH-SNN with existing sparse SNNs training methods including Grad R (Chen et al., 2021), SD-SNN (Han et al., 2025a) and DPAP (Han et al., 2025b), using the same network architectures for fair comparison. In addition, we have conducted experiments on the Spikformer (Zhou et al., 2022) architecture to further verify our methods. It is worth noting that none of the compared methods have been evaluated on the Spikformer. Detailed experimental settings are provided in Appendix A.5 and reproducibility statement is provided in Appendix A.10. The experimental results are summarized in Table 1 and Figure 2.

**Performance on Non-Spiking Datasets.** On the MNIST dataset with the two-layer fully connected (2FC) architecture, CH-SNN attains a 97.75% sparsity while improving accuracy by 0.16% over the FC baseline. Compared to the state-of-the-art method DPAP, CH-SNN not only increases sparsity by approximately 20% but also achieves a performance gain of 0.23%. With the 2CONV2FC architecture, CH-SNN realizes 93.91% sparsity with a 0.16% improvement in accuracy. These results demonstrate that CH-SNN delivers both the highest level of sparsity and the most significant performance gains among all compared methods. Furthermore, when applied to the Spikformer architecture, CH-SNN achieves an 81.72% sparsity with a slight performance improvement of 0.02%. On the CIFAR-10 dataset, CH-SNN again achieves the highest level of sparsity—74.62%—with only a minimal accuracy drop of 0.14%. Similarly, when applied to Spikformer, it maintains a sparsity of 82.21% with a negligible performance degradation of 0.10%. On the CIFAR-100 dataset, although the performance improvement of CH-SNN is marginally lower than that of SD-SNN, it increases sparsity by nearly 38%. Additionally, with the Spikformer architecture, CH-SNN provides a 0.75% performance improvement at 82.11% sparsity.

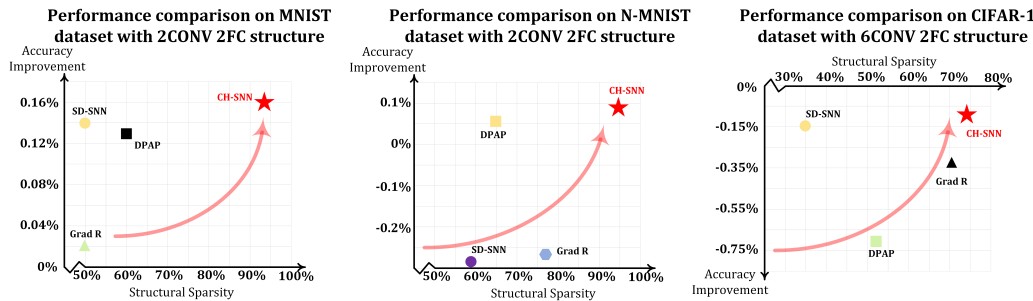

Figure 2: Performance comparison of different methods on MNIST, N-MNIST and CIFAR-10. We plot the performance of different sparse SNN training methods with structural sparsity on the x-axis and accuracy improvement on the y-axis. The plot clearly shows that CH-SNN achieves the highest level of sparsity alongside the greatest improvement in accuracy.

Table 1: Performance comparison of different methods on CIFAR-10, CIFAR-100, MNIST, N-MNIST, CIFAR10-DVS and DVS-Gesture datasets. The gray section indicates the performance of CH-SNN. For each dataset, we have bolded the method with the highest accuracy improvement and the one with the highest sparsity. We exclude Spikformer from our comparison here.

| Dataset | Method | Network | Link sparsity | Acc. | Accuracy improvement |
|---|---|---|---|---|---|
| CIFAR-10 | Grad R | 6Conv 2FC | 71.59% | 92.54% | –0.30% |
| | DPAP | 6Conv 2FC | 50.80% | 93.83% | –0.71% |
| | SD-SNN | 6Conv 2FC | 35.57% | 94.59% | –0.15% |
| | **CH-SNN** | **6Conv 2FC** | **74.62%** | **94.60%** | **–0.14%** |
| | CH-SNN | Spikformer | 82.21% | 94.26% | –0.10% |
| CIFAR-100 | **SD-SNN** | **6Conv 2FC** | **36.94%** | **75.33%** | **+3.27%** |
| | **CH-SNN** | **6Conv 2FC** | **74.45%** | **75.22%** | **+3.16%** |
| | CH-SNN | Spikformer | 82.11% | 76.23% | +0.75% |
| MNIST | Grad R | 2FC | 74.29% | 98.59% | –0.33% |
| | DPAP | 2FC | 77.36% | 98.74% | –0.07% |
| | SD-SNN | 2FC | 45.86% | 98.90% | +0.09% |
| | **CH-SNN** | **2FC** | **97.75%** | **98.97%** | **+0.16%** |
| | Grad R | 2Conv 2FC | 49.16% | 99.37% | +0.02% |
| | DPAP | 2Conv 2FC | 61.25% | 99.59% | +0.13% |
| | SD-SNN | 2Conv 2FC | 49.83% | 99.51% | +0.14% |
| | **CH-SNN** | **2Conv 2FC** | **93.91%** | **99.53%** | **+0.16%** |
| | CH-SNN | Spikformer | 81.72% | 99.73% | +0.02% |
| N-MNIST | Grad R | 2Conv 2FC | 75.00% | 98.56% | –0.27% |
| | DPAP | 2Conv 2FC | 63.95% | 99.59% | +0.06% |
| | SD-SNN | 2Conv 2FC | 58.62% | 98.78% | –0.29% |
| | **CH-SNN** | **2Conv 2FC** | **94.73%** | **99.15%** | **+0.08%** |
| | CH-SNN | Spikformer | 85.74% | 99.45% | +0.10% |
| CIFAR10-DVS | **CH-SNN** | **6Conv 2FC** | **84.34%** | **72.00%** | **+1.50%** |
| | CH-SNN | Spikformer | 85.37% | 70.60% | +0.40% |
| DVS-Gesture | Deep R | 2Conv 2FC | 75.00% | 81.23% | –2.89% |
| | **Grad R** | **2Conv 2FC** | **75.00%** | **91.95%** | **+7.83%** |
| | SD-SNN | 2Conv 2FC | 61.10% | 96.21% | +1.14% |
| | **CH-SNN** | **2Conv 2FC** | **94.73%** | **95.45%** | **+0.38%** |
| | CH-SNN | Spikformer | 82.25% | 93.56% | +1.14% |

**Performance on Spiking Datasets.** On the N-MNIST dataset, CH-SNN attains a 0.08% performance improvement at 94.73% sparsity, outperforming the FC network. On the DVS-Gesture dataset, although CH-SNN's accuracy gain is lower than those of SD-SNN and Grad R, it still demonstrates 95.45% accuracy with 0.38% improvement compared to the FC baseline and reaches significantly higher sparsity level (94.73%) than all other compared methods. On the CIFAR10-DVS dataset, CH-SNN exhibits an accuracy improvement of 1.50% at 84.34% sparsity. Since the CH-SNN framework removes inactive neurons, the sparse network with CH-SNN achieves higher node sparsity compared to other methods. Detailed results are provided in Table 9 of the appendix A.8.

## 4.2 EXPERIMENTS ON HARDWARE-FRIENDLY ALGORITHMS S-TP

Hardware-friendly algorithm S-TP has been realized in a chip ANP-I (Zhang et al., 2024a), which is a low-power neuromorphic processor for edge-side AI applications. To verify our methods, we apply CH-SNN to S-TP. Detailed experimental configurations and network architectures are provided in Appendix A.5. We have conducted comprehensive experiments to evaluate the performance of CH-SNN in terms of accuracy and energy efficiency. The results are shown in Table 2 and Figure 3.

**Accuracy Analysis.** At an average sparsity of 96.4%, the sparse network with CH-SNN achieves the comparable accuracy of the FC network without CH-SNN on all four datasets. Notably, on the DVS-Gesture dataset, sparse network attains a 2.27% improvement in accuracy at 98.84% sparsity.

For the N-MNIST dataset, although the accuracy of the sparse network is slightly lower than that of the FC network, it successfully prunes nearly half of the nodes (41.90% node sparsity) with only a minimal accuracy drop of 0.18%.

Table 2: Performance and energy consumption of CH-SNN on MNIST, DVS-Gesture, N-MNIST and CIFAR-10 datasets (S-TP). For each dataset, the first row shows the performance of the fully connected network, and the second row shows that of the sparse network with CH-SNN. A 3FC architecture is consistently employed across all experiments, with details provided in the appendix A.5.

| Dataset | Spike count | SOPs | Firing rate | Link sparsity | Node sparsity | Acc. | Energy |
|---|---|---|---|---|---|---|---|
| **MNIST** | $6.1 \times 10^8$ | $6.3 \times 10^{11}$ | 31.22% | 0% | 0% | 97.29% | 948mJ |
| | $\mathbf{3.3 \times 10^8}$ | $\mathbf{3.2 \times 10^{10}}$ | **16.83%** | **94.59%** | **23.47%** | **97.56%** | **48mJ** |
| **DVS-Gesture** | $2.2 \times 10^7$ | $5.2 \times 10^{10}$ | 4.02% | 0% | 0% | 89.02% | 78mJ |
| | $\mathbf{1.3 \times 10^7}$ | $\mathbf{5.0 \times 10^8}$ | **2.45%** | **98.84%** | **12.30%** | **91.29%** | **0.8mJ** |
| **N-MNIST** | $2.5 \times 10^8$ | $1.4 \times 10^{11}$ | 10.42% | 0% | 0% | 96.38% | 216mJ |
| | $\mathbf{1.2 \times 10^8}$ | $\mathbf{2.9 \times 10^9}$ | **4.72 %** | **98.46%** | **41.90%** | **96.20%** | **4.4mJ** |
| **CIFAR-10** | $6.3 \times 10^7$ | $2.8 \times 10^{10}$ | 30.70% | 0% | 0% | 80.67% | 41mJ |
| | $\mathbf{1.9 \times 10^7}$ | $\mathbf{4.8 \times 10^8}$ | **9.28%** | **93.78%** | **0%** | **82.84%** | **0.7mJ** |

**Energy Analysis.** We evaluate the energy efficiency in terms of the average firing rate, total spike count, and the number of Synaptic Operations (SOPs). The measured energy consumption of ANP-I chip is 1.5 pJ/SOP (Zhang et al., 2024a), which is regarded as a baseline. We calculate the total energy consumption by multiplying this baseline by the total SOPs. As illustrated in Figure 3, on the DVS-Gesture dataset, the CH-SNN with sparse connectivity consumes 97.5 times less energy than its fully connected counterpart. Furthermore, it yields an average reduction in energy consumption of 55 times and a 10.77% decrease in the average firing rate across the four datasets.

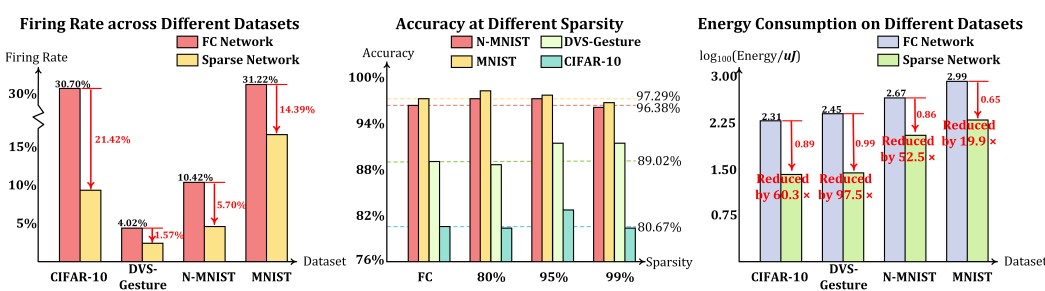

Figure 3: Experimental results of CH-SNN on hardware-friendly algorithm S-TP. A comparison of firing rates between the sparse network with CH-SNN and the FC network across four datasets is presented in the left plot. The middle plot provides an accuracy comparison among the FC network and sparse networks at sparsity levels of 80%, 95%, and 99%. The right plot displays the energy consumption comparison, with the vertical axis on a base-100 logarithmic scale. A scale difference of 0.99 signifies that the energy consumption of the FC network is 97.5 ($100^{0.99}$) times greater than that of the sparse network.

**Supplementary Experiments.** We perform ablation studies to verify the SSCTI and SSWI methods, with results in Appendix A.6. Besides, sensitivity analyses are conducted on critical hyperparameters—including learning rate, batch size and pruning ratio—as summarized in Appendix A.7. We also conducted a robustness analysis and details are provided in Appendix A.9. Details of large language model usage in the writing process can be found in the Appendix A.13.

## 5 CONCLUSION

To address the challenge in achieving high levels of structural connection sparsity while maintaining performance comparable to that of fully connected networks, this paper presents CH-SNN, a four-stage dynamic sparse training framework for learning ultra-sparse spiking neural networks. The framework comprises: (1) sparse topology initialization, leveraging input correlation to initialize the network structure; (2) sparse weight initialization, which incorporates temporal activation sparsity, structural connection sparsity and neuronal threshold information of SNNs to initialize the weights of SNNs within a sparse network structure. (3) network pruning based on a removal score, combined with the removal of inactive neurons to improve information flow, and (4) network regrowth using the CH3-L3 score with a probabilistic strategy. These stages enable CH-SNN to achieve sparsification across all linear layers and enables effective training at ultra-high levels of sparsity conditions. Experimental results demonstrate that CH-SNN outperforms existing sparse SNNs training methods, achieving 97.75% sparsity on MNIST with a 0.16% accuracy improvement over the FC baseline. In addition, it realizes 98.84% sparsity with a 2.27% performance gain over the FC network while improving energy efficiency by approximately 97.5×. In summary, CH-SNN achieves performance comparable to FC networks even at ultra-high levels of sparsity, offering a promising solution for implementing edge AI on neuromorphic hardware.

### ACKNOWLEDGMENTS

This work is supported by National Natural Science Foundation of China (No. 62574117 and 62334014).

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

# A  APPENDIX

## A.1  SPARSE SPIKE WEIGHT INITIALIZATION

For the sparse spiking neural network, we first introduce the sparse connection matrix $C \in \{0,1\}^{m \times n}$, where 1 represents a connection exists and 0 represents no connection exists. For a sparse network, we have:

$$\boldsymbol{y}^{(l)} = \boldsymbol{C}^{(l)} \odot \boldsymbol{W}^{(l)} \boldsymbol{x}^{(l)} + \boldsymbol{b}(l) \tag{6}$$

where $\odot$ represents the Hadamard product, $\boldsymbol{W}^{(l)} \in \mathbb{R}^{m \times n}$ is the weight matrix, $\boldsymbol{x}^{(l)} \in \mathbb{R}^{n \times 1}$ denotes the input vector, $\boldsymbol{b}^{(l)} \in \mathbb{R}^{m \times 1}$ is the bias vector, $\boldsymbol{y}^{(l)} \in \mathbb{R}^{m \times 1}$ stands for the output vector, and $l$ represents the layer index. Next we assume that bias is 0 and the number of network layers is $L$, For the $i-$th element $y_i^{(l)}$ of the output vector $\boldsymbol{y}^{(l)}$, we discuss its variance:

$$Var[y_i^{(l)}] = Var[\sum_{j=1}^{n} C_{ij}^{(l)} W_{ij}^{(l)} x_j^{(l)}] \tag{7}$$

$C$, $W$ and $x$ are independent of each other, All elements in the matrices $W$,$C$,$x$ are independently and identically distributed, and the three matrices have different distributions. Then the variance of $y_i^{(l)}$ can be expressed as follows:

$$Var[y_i^{(l)}] = n[(Var[C_{ij}] + \mu_C^2)(Var[W_{ij}] + \mu_W^2)(Var[x_j] + \mu_x^2) - \mu_C^2 \mu_W^2 \mu_x^2] \tag{8}$$

where $\mu$ is the mean. For the elements of the sparse connection matrix $C$, we assume that it follows the Bernoulli distribution, then we have:

$$C_{ij} = \begin{cases} 1, \ p = 1 - S_s \\ 0, \ p = S_s \end{cases}, \quad \mu_C = 1 - S_s, \quad Var[C_{ij}] = S_s(1 - S_s) \tag{9}$$

where $S_s$ is the structural connection sparsity. Based on previous work (He et al., 2015), we similarly define $W_{ij}$ to follow a zero-mean Gaussian distribution, which means $\mu_W = 0$. The Equation (8) can be changed to:

$$Var[y_i^{(l)}] = n(1 - S_s)(Var[W_{ij}])(Var[x_j] + \mu_x^2) \tag{10}$$

We expect to maintain the same variance of the input feature across layers, which means $Var[y_i^{(l)}] = Var[y_i^{(l-1)}]$. For the input layer ($l = 1$), it can be expressed as $Var[y_i^{(1)}] = Var[x_i]$, because there is no activation function applied to the input feature. The input of the spiking neural network is 1 or 0, therefore, we can assume that $x_i$ follows the Bernoulli distribution with parameter $p = S_t$. We can get the variance of the $W_{ij}$ as follows:

$$Var[W_{ij}^{(1)}] = \frac{S_t}{n(1 - S_s)} \tag{11}$$

where $S_t$ is the temporal activation sparsity, and $n$ is the dimension of the input. For the rest of sparsely connected layers, $y_i^{(l)}$ follows a distribution with zero mean and symmetric about zero, since $W_{ij}^{(l)}$ follows a zero-mean Gaussian distribution and the input of every layer is zero or one,

which will not affect the symmetry of $W_{ij}^{(l)}$. To simplify the analysis, we assume that $y_i^{(l)}$ also follows a zero-mean Gaussian distribution. In the spiking neural network, the activation function is not ReLU, but rather the step function, which can be expressed as:

$$x_i^{(l)} = U(y_i^{(l-1)} + V_{mem} - \theta) = \begin{cases} 1, & y_i^{(l-1)} + V_{mem} \geq \theta \\ 0, & y_i^{(l-1)} + V_{mem} < \theta \end{cases} \quad (12)$$

Where $\theta$ is the threshold of the spiking neural network. We approximate that $V_{mem}$ and $y_i^{(l-1)}$ follow the same distribution. Therefore, Equation (12) can be expressed as follows:

$$x_i^{(l)} = U(2y_i^{(l-1)} - \theta) = \begin{cases} 1, & y^{(l-1)} \geq \theta/2 \\ 0, & y^{(l-1)} < \theta/2 \end{cases} \quad (13)$$

We expect to find a relationship between $Var[x_j^{(l)}] + \mu_x^2$ and $Var[y_j^{(l-1)}]$ that will simplify Equation (10). Based on Equation (13), we have:

$$Var[x_j^{(l)}] + \mu_x^2 = \int_{\theta/2}^{+\infty} \frac{1}{\sqrt{2\pi Var[y_j^{(l-1)}]}} e^{-x^2/2Var[y_j^{(l-1)}]} dx = 1 - \Phi(\theta/2\sqrt{Var[y_j^{(l-1)}]}) \quad (14)$$

where $\Phi(\cdot)$ is the cumulative distribution function of the standard normal distribution. We expand Equation (14) in a Taylor series around $\theta/2$ with $Var[y_j^{(l-1)}]$ as the variable and discard the higher-order terms:

$$Var[x_j^{(l)}] + \mu_x^2 \approx 1 - \Phi(1) + \frac{2}{\sqrt{2\pi\theta^2}} e^{(-\frac{1}{2})}(Var[y_j^{(l-1)}] - \frac{\theta^2}{4}) \approx \frac{\sqrt{2}e^{-1/2}}{\sqrt{\pi}\theta^2} Var[y_j^{(l-1)}] \quad (15)$$

Next, we substitute Equation (15) into Equation (10), obtaining the following result:

$$Var[W_{ij}^{(l)}] = \frac{\theta^2\sqrt{\pi}}{\sqrt{2}e^{-1/2}n(1-S_s)}, (1 < l < L) \quad (16)$$

Finally, we assume that the output layer is fully connected, which means $S_s = 0$, so we can obtain the following result:

$$Var[W_{ij}^{(L)}] = \frac{\theta^2\sqrt{\pi}}{\sqrt{2}e^{-1/2}n} \quad (17)$$

In summary, we can summarize the Sparse Spike Weight Magnitude Initialization (SSWI) as follows:

$$SSWI(W_{ij}^{(l)}) \sim \mathcal{N}(0, \sigma^2), \quad \sigma^2 = \begin{cases} \dfrac{S_t}{n(1-S_s)}, & (l = 1) \\[3mm] \dfrac{\theta^2\sqrt{\pi}}{\sqrt{2}e^{-1/2}n(1-S_s)}, & (1 < l < L) \\[3mm] \dfrac{\theta^2\sqrt{\pi}}{\sqrt{2}e^{-1/2}n}, & (l = L) \end{cases} \quad (18)$$

## A.2 EXPLANATION OF CH3-L3

To illustrate the application of CH3-L3 for link prediction, consider a hypothetical network containing nodes $u$ and $v$ that are not directly connected. The CH3-L3 method evaluates potential links by analyzing all length-3 paths between $u$ and $v$ and incorporating the local community connectivity of intermediate nodes.

We provide an illustrative example in Figure 4. Red nodes represent seed nodes between which no direct connection is currently observed, but which have the potential to regenerate a link. Green nodes denote common neighbors—nodes directly connected to both seed nodes. Red edges indicate connections between common neighbors, referred to as internal Local Community Links (iLCL). Blue nodes represent external nodes outside the set of seed nodes and their common neighbors. Blue edges correspond to connections between common neighbors and these external nodes, termed external Local Community Links (eLCL). The calculation process of CH3-L3 is as follows:

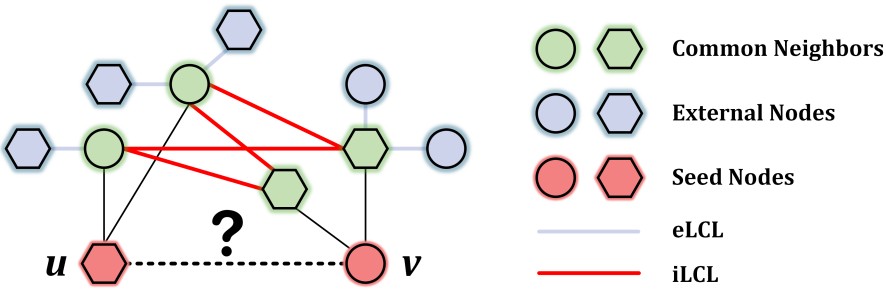

Figure 4: Example of link prediction using CH3-L3.

$$\textbf{CH3-L3}(u,v) = \sum_{z_1,z_2 \in l3(u,v)} \frac{1}{\sqrt{(1+de_{z_1}) \times (1+de_{z_2})}} \qquad (19)$$

where $u$ and $v$ denote two seed nodes, and $z_1$, $z_2$ represent two common neighbor nodes along a path of length 3 between $u$ and $v$. The terms $de_{z_1}$ and $de_{z_2}$ correspond to the number of eLCL associated with node $z_1$ and $z_2$, respectively. The CH3-L3 score is computed by summing the contributions from all length-3 paths between $u$ and $v$. Here, adding 1 to $de_{z_1}$ and $de_{z_2}$ prevents division by zero and ensures numerical stability when no external links are present.

### A.2.1 GUARANTEEING TRAINING STABILITY VIA THE CH3-L3

The CH3-L3 regrowth score is derived from the Cannistraci-Hebb theory of local-community organization (Cannistraci, 2018b; Zhang et al., 2024d; Zhao et al., 2025). The term $d_{ez}$ in Equation 19 (the external degree of a common neighbor $z$) acts as a penalty term. It inherently discourages the formation of new links that would connect to nodes that are already highly connected outside their immediate local community. This design naturally prevents the mechanism from getting stuck in redundant loops of adding and removing the same connections. It preferentially strengthens connections within structurally isolated communities, which is a stable, self-reinforcing process. It is a gradient-free, topology-driven predictive network automata (Zhang et al., 2025; Zhao et al., 2025) that guides the network towards a sparse, hub-and-community structure (Zhang et al., 2024d).

While CH3-L3 is inherently stable, we proactively introduced an early-stop mechanism in the CH-SNN framework to explicitly monitor and halt any potential redundancy. We track the overlap rate between the set of links removed and the set of links regrown in each topological update cycle. Once this overlap rate exceeds a high threshold (e.g., 95%), it signals that the network topology has stabilized. At this point, we permanently stop the topological evolution (pruning and regrowth) for that layer and focus solely on weight update.

Furthermore, our framework includes a chain removal, as shown in section 3.2.3, that permanently removes inactive neurons. Since these neurons cannot attract new connections via CH3-L3, their removal prevents structural dead-ends and further contributes to overall training stability and efficiency.

Our extensive experiments across multiple datasets and architectures, where CH-SNN consistently converges and outperforms baselines, serve as empirical validation of this stable behavior. Finally, the previous studies on dynamic sparse training (DST) never raised a concern of stable convergence or meaningful structure learning because the DST methodology selects which link should be removed during the training by using the weight update, this ensures a convergence of the model towards meaningful structures (Zhang et al., 2024d; Evci et al., 2019; Zhang et al., 2024c) even with a random predictor such as SET (Mocanu et al., 2018).

### A.2.2 BIOLOGICAL INTERPRETABILITY OF CH3-L3

The CH3-L3 mechanism operationalizes Hebbian principles at a network-structural level, embodying the concept that "neurons that fire together wire together" (Hebb, 1949). Specifically, it identifies

pairs of neurons that exhibit structural correlations, such as sharing common neighbors or belonging to the same local community, and predicts the formation of new links between them. This process mirrors synaptic turnover and rewiring observed in biological neural circuits, where connections are dynamically formed or strengthened based on functional relatedness.

Although CH3-L3 does not explicitly model precise spike timing, it captures the topological outcomes of STDP-like plasticity. CH3-L3 leverages these structural signatures, derived from local community detection, to guide link regrowth, effectively simulating a topology-driven form of plasticity. This approach is analogous to how STDP reinforces connections within functionally related neuronal assemblies, promoting sparse yet efficient network dynamics. In dynamic sparse networks, such regrowth ensures adaptability while maintaining biological realism, as it avoids random reconnections and prioritizes structurally plausible ones.

In summary, while CH3-L3 is not a direct biophysical simulation of STDP, it provides a functionally equivalent and biologically interpretable mechanism for synaptic rewiring, enhancing the model's relevance to neuroscience applications.

### A.3 DETAILS OF NETWORK PRUNING AND NETWORK REGROWTH

During dynamic sparse training, we generate a sparse connectivity matrix $C$, where $C_{ij} = 1$ indicates the presence of a link between nodes $i$ and $j$, and $C_{ij} = 0$ indicates no link between the nodes. After each training epoch, a process of network pruning and network regrowth is performed. Correspondingly, the sparse connectivity matrix $C$ is updated.

#### A.3.1 LINK REMOVAL

During the link removal phase, we calculate the Link Removal Score ($LRS$) for the existing links (where $C_{ij} = 1$). The $LRS_{ij}$ is computed as shown in Equation (20):

$$LRS_{ij} = (\frac{|W_{ij}|}{1 + \sum_i |W_{ij}|} + \frac{|W_{ij}|}{1 + \sum_j |W_{ij}|})^{\frac{\delta}{1-\delta}} \tag{20}$$

The parameter $\delta$ controls the sampling distribution. When $\delta = 0$, it means the $LRS$ is identical for all links. In this scenario, the corresponding sampling method is random sampling, where links are randomly selected and removed. When $\delta = 1$, the sampling becomes deterministic, and links are removed directly based on their $LRS$ values. When $\delta = 0.5$, it means sampling from a multinomial distribution based on the $LRS$ values. In this situation, we calculate the link removal probability $p_{ij}^{(removal)}$ using the $LRS_{ij}$, as shown in Equation (21):

$$p_{ij,}^{(removal)} = \frac{LRS_{ij}}{\sum_{ij} LRS_{ij}} \qquad i, j \in \{a, b \mid C_{ab} = 1\} \tag{21}$$

Subsequently, based on the link removal probabilities, we remove a certain proportion ($\zeta$) of links as specified in Equation (22), completing the link removal process.

$$C_{ij} = \begin{cases} 1 & 1 - p_{ij}^{(removal)} \\ 0 & p_{ij}^{(removal)} \end{cases} \qquad i, j \in \{a, b \mid C_{ab} = 1\} \tag{22}$$

#### A.3.2 LINK REGROWTH

During link regrowth, we compute the link regrowth score (**CH3-L3**$(u, v)$) for nonexistent links (where $C_{uv} = 0$), as formulated in Equation (23).

$$\textbf{CH3-L3}(u, v) = \sum_{z_1, z_2 \in l3(u,v)} \frac{1}{\sqrt{(1 + de_{z_1}) \times (1 + de_{z_2})}}^{\frac{\delta}{1-\delta}} \tag{23}$$

Consistent with the link removal, the parameter $\delta$ controls the sampling distribution. $\delta = 0$ represents random sampling, $\delta = 1$ indicates that links will be directly regrown based on the

**CH3-L3**$(u, v)$, and $\delta = 0.5$ signifies sampling from a multinomial distribution based on the **CH3-L3**$(u, v)$. Similarly, we compute the link regrowth probability $p_{uv}^{(regrowth)}$, as shown in Equation (24).

$$p_{uv}^{(regrowth)} = \frac{\textbf{CH3-L3}(u, v)}{\sum_{uv} \textbf{CH3-L3}(u, v)} \qquad u, v \in \{a, b \mid C_{ab} = 0\} \tag{24}$$

Following the regrowth probabilities, the regrowth of links can be completed as shown in Equation (25). The number of regrowth links remains consistent with the number of pruned links, thereby maintaining the pre-defined overall network sparsity.

$$C_{uv} = \begin{cases} 1 & p_{uv}^{(regrowth)} \\ 0 & 1 - p_{uv}^{(regrowth)} \end{cases} \qquad u, v \in \{a, b \mid C_{ab} = 0\} \tag{25}$$

To evaluate the impact of different regrowth sampling distributions on model accuracy, we conducted comprehensive experiments. As shown in Table 3, the highest accuracy is achieved when $\delta = 0.5$. Consequently, for all experimental results reported in this paper, we set $\delta = 0.5$, which means we sample regrowth links from a multinomial distribution.

Table 3: Sensitivity experiment of $\delta$.

| Dataset | $\delta = 1$ | $\delta = 0.5$ | $\delta = 0$ |
|---|---|---|---|
| N-MNIST | 96.30% | 97.21% | 96.63% |
| DVS-Gesture | 89.02% | 91.29% | 90.53% |
| MNIST | 98.28% | 98.40% | 98.30% |
| CIFAR-10 | 78.87% | 82.84% | 77.68% |

### A.4 Details of Sparse Spike Correlated Topological Initialization

To enhance the performance of the link predictor, we propose the Sparse Spike Correlated Topological Initialization (SSCTI), a method for initializing the topology of layers that interact directly with input features.

Assuming we have $N$ samples with $M$ input features and a timestep of $T$, the SSCTI procedure consists of the following four steps, as illustrated in Figure 5. **(1) Temporal Unfolding and Stacking.** We treat each timestep as an independent sample and stack them along the feature dimension, resulting in a data matrix of $\mathbb{R}^{M \times NT}$ for subsequent processing. **(2) Feature Correlation Calculation.** Based on the $\mathbb{R}^{M \times NT}$ matrix, we compute the Pearson's phi coefficient between every pair of features, constructing a feature correlation matrix of $\mathbb{R}^{M \times M}$. **(3) Top-K Sparsification and Adjacency Matrix Construction.** We retain only the top-$(1 - S_s)$ values in the correlation matrix (where $S_s$ denotes the structural connection sparsity) and set them to 1, forming an adjacency matrix of $\mathbb{R}^{M \times M}$ that defines the initial topological structure of the layer. When the hidden layer dimension matches the input dimension ($M$), connections exist only where the adjacency matrix contains 1. If the hidden layer is larger (e.g., $2M$), the adjacency matrix is tiled or repeated accordingly (e.g., to $\mathbb{R}^{M \times 2M}$), allowing flexible scaling of hidden dimensions. The same process is applied to initialize topological structures between hidden layers. **(4) Sparse Topological Initialization.** Finally, we initialize the network topology based on the adjacency matrix, preserving only the connections indicated by 1's in the matrix.

### A.5 Experimental setup

To evaluate the performance of CH-SNN, we conduct extensive experiments on multiple datasets, including MNIST (Deng, 2012), CIFAR10 (Krizhevsky, 2009), CIFAR100 (Krizhevsky, 2009), N-MNIST (Orchard et al., 2015), CIFAR10-DVS (Li et al., 2017) and DVS-Gesture (Amir et al., 2017). Both the MNIST and N-MNIST datasets contain 10 classes of handwritten numbers (0–9), each consisting of 60,000 training samples and 10,000 test samples. DVS-Gesture comprises 11

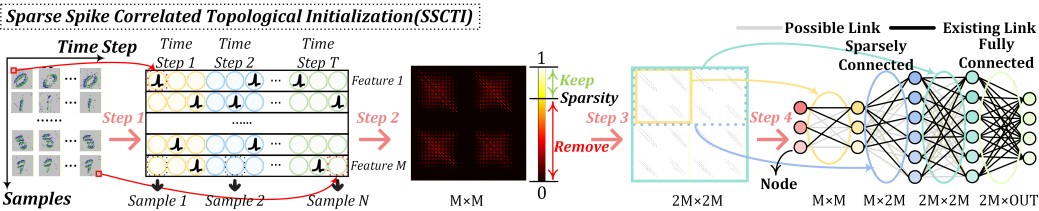

Figure 5: An example of how to construct the SSCTI on the N-MNIST dataset

types of gestures captured using an event-based camera. For these three datasets, we adopt the following network architecture: Input → **15** (Channel Count)**Conv** (Layer Type)**3×3**(Kernel Size) → AvgPool2×2 → 40Conv3×3 → AvgPool2×2 → Flatten → 300 → Classes. For MNIST, a two-layer fully connected network is used: 784 → 1568 → 10. Since CIFAR10 and CIFAR100 share a similar data format, we employ the same network structure for both: Input → [128Conv3×3]×2 → MaxPool2×2 → [256Conv3×3]×2 → MaxPool2×2 → 512Conv3×3 → Flatten → 512×8×8 → 512 → 10. All networks are configured with a time step of 8. Non-Spiking datasets (CIFAR10, CIFAR100, MNIST) are encoded using direct encoding. For all experiments, we adopt the standard Spikformer architecture (Zhou et al., 2022), configured with 8 encoder layers, a hidden dimension of 512, 8 attention heads, and a timestep of 4. A uniform sparsity of 99% is applied to all linear layers (except for the output layer). Weight updates are performed via surrogate gradient methods. Within each epoch, after completing weight training, CH-SNN performs pruning and regeneration according to the pruning ratio $\zeta$ followed by testing. The value of $\zeta$ decays cosine-annealingly to zero over the course of training, which can be expressed as follows:

$$\zeta = \frac{\zeta}{2} \times (1 + \cos(\frac{\pi \times epoch}{total\ epochs}))$$

(26)

For the S-TP algorithm, we perform experiments on the CIFAR-10, MNIST, N-MNIST and DVS-Gesture datasets using a network structure defined as Input-2×Input-2×Input-Classes, with input sizes of 578 for N-MNIST, 784 for MNIST, 512 for CIFAR-10, and 2048 for DVS-Gesture. Throughout training, we set the learning rate to 0.0001, the dynamic pruning ratio $\zeta$ to 0.35, the batch size to 100, the target window size to 4, the number of epochs to 100, and the neuronal firing threshold to 4.

## A.6 ABLATION EXPERIMENT

We have conducted ablation studies to validate the effectiveness of our proposed SSCTI and SSWI. The results are summarized in Table 4. When both SSCTI and SSWI are removed, the model fails to converge in most cases. Removing either SSCTI or SSWI individually leads to varying degrees of performance degradation. Notably, with a high level of sparsity of 99%, the model becomes unstable and fails to train when SSWI is ablated. These ablation results demonstrate the critical importance and effectiveness of both SSCTI and SSWI in maintaining network performance under extreme sparsity.

## A.7 SENSITIVITY TEST

We have conducted a sensitivity analysis of the hyperparameters in CH-SNN, focusing primarily on the learning rate, batch size, dynamic pruning ratio, and static sparsity rate, to evaluate the model's performance under variations in these parameters.

**Learning Rate (LR).** We train CH-SNN using different learning rates (0.01, 0.005, 0.001, 0.0005, 0.0001) and record its performance, as summarized in Table 5. The results indicate that as the learning rate increases, the model exhibits a noticeable decline in performance. Through further analysis, we conclude that this performance degradation stems from the S-TP algorithm: during weight updates, an excessively large learning rate leads to oversized training steps, preventing convergence to an optimal solution. This is validated by conducting a learning rate sensitivity experiment on a fully connected network, where a similar performance drop is observed, as shown in Table 5.

Table 4: Ablation Study on SSCTI and SSWI (When SSCTI is ablated, we employ random structural initialization; when SSWI is removed, we use Kaiming initialization).

| Dataset | SSCTI | SSWI | 99% Sparsity | 95% Sparsity | 80% Sparsity | 70% Sparsity |
|---|---|---|---|---|---|---|
| **N-MNIST** | | | 9.80% | 9.80% | 9.80% | 91.99% |
| | ✓ | | 9.80% | 88.82% | 96.95% | 97.12% |
| | | ✓ | 89.61% | 94.74% | 96.53% | 96.87% |
| | ✓ | ✓ | **96.20%** | **97.21%** | **97.29%** | **97.22%** |
| **MNIST** | | | 9.80% | 9.80% | 97.23% | 97.78% |
| | ✓ | | 70.41% | 96.57% | 97.57% | 97.57% |
| | | ✓ | 78.09% | 96.81% | 97.88% | 97.91% |
| | ✓ | ✓ | **96.88%** | **97.56%** | **98.11%** | **98.00%** |
| **DVS-Gesture** | | | 9.09% | 9.09% | 9.09% | 9.09% |
| | ✓ | | 9.09% | 9.09% | 78.79% | 86.74% |
| | | ✓ | 76.52% | 87.12% | 87.50% | 86.74% |
| | ✓ | ✓ | **91.29%** | **91.29%** | **88.64%** | **89.02%** |
| **CIFAR-10** | | | 10.00% | 57.35% | 78.89% | 77.95% |
| | ✓ | | 34.55% | 76.83% | 80.01% | 78.19% |
| | | ✓ | 77.40% | 81.62% | 80.62% | 80.65% |
| | ✓ | ✓ | **78.94%** | **82.84%** | **81.73%** | **81.42%** |

Table 5: Learning rate sensitivity experiment.

| Dataset | lr-0.01 | lr-0.005 | lr-0.001 | lr-0.0005 | lr-0.0001 |
|---|---|---|---|---|---|
| N-MNIST | 83.08% | 85.59% | 95.84% | 96.87% | 96.91% |
| DVS-Gesture | 65.91% | 75.76% | 89.77% | 89.77% | 89.77% |
| MNIST | 67.18% | 85.83% | 94.98% | 96.19% | 97.98% |
| CIFAR-10 | 78.74% | 79.96% | 82.84% | 82.73% | 82.54% |
| N-MNIST(FC) | 69.65% | 80.90% | 88.16% | 96.31% | 96.38% |
| DVS-Gesture(FC) | 68.56% | 68.18% | 85.98% | 88.64% | 89.02% |
| MNIST(FC) | 79.90% | 84.58% | 92.43% | 94.12% | 97.29% |
| CIFAR-10(FC) | 59.86% | 55.49% | 80.14% | 81.92% | 80.67% |

**Batch Size (BS).** We employ different batch sizes to train CH-SNN, and the experimental results presented in Table 6 show that variations in batch size did not cause significant changes in its performance.

Table 6: Batch size sensitivity experiment.

| Dataset | BS-16 | BS-32 | BS-50 | BS-64 | BS-100 |
|---|---|---|---|---|---|
| N-MNIST | 96.88% | 96.86% | 96.85% | 97.00% | 96.91% |
| DVS-Gesture | 89.77% | 89.39% | 89.02% | 89.39% | 89.77% |
| MNIST | 98.08% | 98.09% | 98.11% | 98.07% | 97.98% |
| CIFAR-10 | 82.64% | 81.95% | 81.88% | 81.74% | 82.84% |

**Dynamic Pruning Ratio ($\zeta$).** To evaluate the impact of different pruning rate strategies on model performance, we first test the dynamic pruning rate strategy as defined in Equation (26). We adjust various decay starting points (0.5, 0.4, 0.3, 0.2, 0.1), and the experimental results are shown in Table 7. It can be observed that CH-SNN exhibits negligible performance variation across different initial pruning rates, demonstrating strong stability in response to changes in the pruning rate. The model consistently achieves strong performance under the predefined sparsity targets.

**Static Pruning Ratio ($\zeta$).** Similarly, we evaluate a static pruning rate strategy. Unlike the dynamic approach, the static pruning rate remains constant at its initial value throughout training. We test

Table 7: Dynamic removal rate sensitivity experiment.

| Dataset | $\zeta$-0.5 | $\zeta$-0.4 | $\zeta$-0.3 | $\zeta$-0.2 | $\zeta$-0.1 |
|---|---|---|---|---|---|
| N-MNIST | 97.14% | 96.91% | 96.96% | 97.04% | 97.10% |
| DVS-Gesture | 89.39% | 90.53% | 89.02% | 88.64% | 89.39% |
| MNIST | 97.98% | 97.98% | 97.98% | 97.98% | 97.98% |
| CIFAR-10 | 82.64% | 82.24% | 82.84% | 83.03% | 82.75% |

multiple starting values for the static pruning rate, and the results are presented in Table 8. CH-SNN shows minimal performance variation across different static pruning rates. It is worth noting that compared to the dynamic pruning strategy, the static approach generally leads to a slight decrease in overall accuracy.

Table 8: Static removal rate sensitivity experiment.

| Dataset | $\zeta$-0.5 | $\zeta$-0.4 | $\zeta$-0.3 | $\zeta$-0.2 | $\zeta$-0.1 |
|---|---|---|---|---|---|
| N-MNIST | 96.90% | 96.87% | 96.84% | 96.81% | 97.10% |
| DVS-Gesture | 88.64% | 89.39% | 89.02% | 89.02% | 88.64% |
| MNIST | 97.98% | 97.98% | 97.98% | 97.98% | 97.98% |
| CIFAR-10 | 82.24% | 82.26% | 82.53% | 82.63% | 82.25% |

## A.8 NODE SPARSITY

In the CH-SNN framework, neurons that are unilaterally or bilaterally disconnected (i.e., without any incoming or outgoing links) are regarded as inactive neurons. Since these inactive neurons lose the ability to transmit information, they may hinder information flow throughout the network. We assume that such inactive neurons are unable to regrow new links during the network regrowth stage. Therefore, during the chain removal step, we permanently remove them from the network. As illustrated in Figure 1 Stage 3, this process enhances node sparsity. We compare CH-SNN with SD-SNN, an existing open-source method, as shown in Table 9.

Table 9: Node sparsity of different methods on CIFAR-10, CIFAR-100, MNIST, N-MNIST and DVS-Gesture datasets.

| Dataset | Method | Network | Node sparsity |
|---|---|---|---|
| **CIFAR-10** | SD-SNN | 6Conv 2FC | 0.28% |
| | **CH-SNN** | **6Conv 2FC** | **1.21%** |
| **CIFAR-100** | SD-SNN | 6Conv 2FC | 0.46% |
| | **CH-SNN** | **6Conv 2FC** | **0.69%** |
| **MNIST** | SD-SNN | 2Conv 2FC | 0.02% |
| | **CH-SNN** | **2Conv 2FC** | **0.04%** |
| | SD-SNN | 2FC | 0.77% |
| | **CH-SNN** | **2FC** | **2.03%** |
| **N-MNIST** | SD-SNN | 2Conv 2FC | 6.32% |
| | **CH-SNN** | **2Conv 2FC** | **8.09%** |
| **DVS-Gesture** | SD-SNN | 2Conv 2FC | 1.28% |
| | **CH-SNN** | **2Conv 2FC** | **6.35%** |

## A.9 ROBUSTNESS ANALYSIS

We train ultra-sparse networks using CH-SNN. In comparison to fully-connected (FC) networks, increasing the structural sparsity leads to a corresponding increase in temporal sparsity. This phenomenon reduces both the number of active spiking neurons and the total spike count. We hypothesize that this loss of information may adversely affect robustness. To test this and evaluate

the robustness of CH-SNN, we conduct experiments where the models, trained on clean data, are exposed to corrupted inputs during testing. The corruption involves three noise types: (1) **Bit-flip noise.** This noise randomly flips 0s to 1s and 1s to 0s. Its destructive nature stems from a dual effect: it corrupts the input by both adding spurious spikes and removing authentic ones. (2) **False-spike noise.** This corruption randomly changes 0s to 1s, which generates extraneous spikes. This directly compromises the timing precision fundamental to SNN operation. (3) **Spike-dropout noise.** This type randomly changes 1s to 0s, thereby dropping genuine spikes. It is designed to emulate spike loss in real neuromorphic hardware.

Table 10: Accuracy on the MNIST dataset with input noise, where P denotes the noise ratio.

| Dataset | Method | $P = 0\%$ | $P = 5\%$ | $P = 10\%$ | $P = 15\%$ | $P = 20\%$ |
|---|---|---|---|---|---|---|
| **Bit-flip** | FC | 97.91% | 97.64% | 95.78% | 83.19% | 68.25% |
| | CH-SNN | 98.21% | 97.86% | 94.75% | 81.19% | 65.03% |
| **False-spike** | FC | 97.91% | 97.74% | 96.50% | 89.97% | 78.39% |
| | CH-SNN | 98.21% | 97.73% | 95.74% | 86.72% | 75.25% |
| **Spike-dropout** | FC | 97.91% | 97.25% | 94.37% | 87.25% | 67.65% |
| | CH-SNN | 98.21% | 97.48% | 93.84% | 83.81% | 66.58% |

Our experimental investigation aimed to evaluate the noise robustness of the proposed CH-SNN. All models in this study were implemented as 3FC networks with a 99% structural sparsity. The testing protocol involved corrupting the input with three noise types across a range of intensities. The noise intensity was controlled by the noise ratio, defined as the proportion of timesteps in the input spike train that are altered. We report the classification accuracy on the MNIST, N-MNIST, CIFAR-10, and DVS-Gesture datasets under these conditions, with the complete results presented in Tables 10, 11, 12, 13.

Table 11: Accuracy on the N-MNIST dataset with input noise, where P denotes the noise ratio.

| Dataset | Method | $P = 0\%$ | $P = 1\%$ | $P = 2\%$ | $P = 3\%$ | $P = 4\%$ |
|---|---|---|---|---|---|---|
| **Bit-flip** | FC | 95.82% | 93.09% | 83.76% | 73.65% | 62.35% |
| | CH-SNN | 96.62% | 94.00% | 83.47% | 74.97% | 65.87% |
| **False-spike** | FC | 95.82% | 93.19% | 84.13% | 74.03% | 63.96% |
| | CH-SNN | 96.62% | 94.08% | 83.89% | 75.7% | 67.32% |
| **Spike-dropout** | FC | 95.82% | 95.85% | 95.77% | 95.78% | 95.78% |
| | CH-SNN | 96.62% | 96.53% | 96.43% | 96.54% | 96.47% |

Table 12: Accuracy on the CIFAR-10 dataset with input noise, where P denotes the noise ratio.

| Dataset | Method | $P = 0\%$ | $P = 1\%$ | $P = 2\%$ | $P = 3\%$ | $P = 4\%$ |
|---|---|---|---|---|---|---|
| **Bit-flip** | FC | 79.23% | 65.55% | 56.16% | 48.15% | 42.63% |
| | CH-SNN | 79.27% | 60.68% | 49.42% | 41.58% | 35.17% |
| **False-spike** | FC | 79.23% | 66.89% | 57.31% | 49.73% | 43.28% |
| | CH-SNN | 79.27% | 60.76% | 49.82% | 41.22% | 34.43% |
| **Spike-dropout** | FC | 79.23% | 79.19% | 79.03% | 78.85% | 78.75% |
| | CH-SNN | 79.27% | 79.14% | 79.03% | 78.88% | 78.62% |

Our results delineate a sharp contrast in robustness against different noise corruptions. Models demonstrate substantial tolerance to spike-dropout noise. With a dropout rate under 5%, accuracy remains largely stable, and we observe a slight performance enhancement on CIFAR-10 and DVS-Gesture. This robustness implies that the random omission of a small number of spikes acts as a mild form of regularization, which is insufficient to alter the overall network output. This finding holds for both sparse and dense networks.

Conversely, bit-flip and false-spike noise cause pronounced performance degradation. Their destructive nature stems from the introduction of spurious spikes, which corrupts the inherent timing-

dependent computation in SNNs. This forces neurons to fire at incorrect timesteps, thereby compromising the integrity of the final decision and resulting in substantial accuracy loss, regardless of network sparsity.

Table 13: Accuracy on the DVS-Gesture dataset with input noise, where P denotes the noise ratio.

| Dataset | Method | $P = 0\%$ | $P = 1\%$ | $P = 2\%$ | $P = 3\%$ | $P = 4\%$ |
|---|---|---|---|---|---|---|
| **Bit-flip** | FC | 87.12% | 83.33% | 79.32% | 78.79% | 57.58% |
| | CH-SNN | 87.12% | 84.85% | 79.92% | 75.38% | 63.26% |
| **False-spike** | FC | 87.12% | 84.47% | 77.65% | 69.32% | 60.98% |
| | CH-SNN | 87.12% | 87.12% | 79.55% | 74.62% | 64.02% |
| **Spike-dropout** | FC | 87.12% | 87.12% | 87.12% | 87.50% | 87.50% |
| | CH-SNN | 87.12% | 87.12% | 87.50% | 87.50% | 87.50% |

The robustness of the model is quantified using the Relative Performance Degradation Rate (RPDR). The RPDR metric is formally defined as Equation 27:

$$\text{RPDR} = \frac{ACC_{clean} - ACC_{noise}}{ACC_{clean}} \tag{27}$$

where $ACC_{clean}$ is the baseline accuracy on unperturbed data, and $ACC_{noise}$ is the accuracy evaluated under a specific noise corruption. This metric is interpreted as follows: a smaller RPDR denotes stronger robustness, as it reflects a smaller relative drop in accuracy. Table 14 presents a comprehensive summary, detailing the RPDR for each noise type alongside the mean RPDR aggregated over all noise conditions.

Table 14: The relative performance degradation rate of the model under the three types of noise.

| Dataset | Method | Bit-flip | False-spike | Spike-dropout | Average |
|---|---|---|---|---|---|
| **MNIST** | FC | **11.95%** | **7.41%** | **11.52%** | **10.29%** |
| | CH-SNN | 13.75% | 9.52% | 13.01% | 12.09% |
| **N-MNIST** | FC | 18.38% | 17.73% | **0.03%** | 12.05% |
| | CH-SNN | **17.64%** | **16.95%** | 0.13% | **11.57%** |
| **CIFAR-10** | FC | **32.95%** | **31.46%** | **0.35%** | **21.59%** |
| | CH-SNN | 41.07% | 41.27% | 0.44% | 27.59% |
| **DVS-Gesture** | FC | 14.19% | 16.09% | –0.22% | 10.02% |
| | CH-SNN | **12.93%** | **12.39%** | **–0.33%** | **8.33%** |

**Analysis on non-spiking datasets (MNIST, CIFAR-10).** On these converted datasets, where static images are encoded into spike trains, the FC baseline exhibits greater robustness on average compared to the CH-SNN. We postulate that the densely and uniformly distributed information in static images aligns better with the FC network's inherent redundancy, granting it higher fault tolerance. The structural sparsity of CH-SNN, while beneficial in other contexts, leads to increased temporal sparsity that offers no robustness advantage here. Consequently, achieving robustness on such tasks would necessitate a design with lower structural sparsity.

**Analysis on native spiking datasets (DVS-Gesture, N-MNIST).** The scenario reverses for native spiking data. When subjected to the most destructive noise types (bit-flip and false-spike), CH-SNN consistently shows a lower performance degradation rate than its FC counterpart. This result underscores that CH-SNN's sparse architecture effectively leverages the inherent sparsity of the data to filter out corrupting noise and protect crucial information, thus validating its stronger robustness for event-based computation.

## A.10 REPRODUCIBILITY STATEMENT

Regarding the experimental results in Table 1, we provide the following two-part clarifications on reproducibility to ensure all comparisons are fair.

**Comparison with other sparse training methods.** For SD-SNN, whose code is publicly available, we faithfully reproduced its sparse training procedure. Specifically, we first trained a dense neural network as the baseline. Under identical experimental conditions, we then applied both SD-SNN and our CH-SNN to obtain their respective sparse networks from this common baseline, guaranteeing a fully fair comparison between the two methods.

For other sparse training methods (e.g., Grad), since their code is not open-source, we did not attempt to reimplement them. Therefore, the results shown in Table 1 for these methods, along with their corresponding dense baseline performances, are directly quoted from their original publications.

**Comparison between dense and sparse networks.** When comparing the dense network with the sparse network trained via CH-SNN, we ensured strict fairness: all training settings and hyperparameters are kept identical, with the only difference being the introduced sparsity.

### A.11 PERFORMANCE ON DIFFERENT TIMESTEPS

The number of timesteps ($T$) is a critical hyperparameter in SNNs, governing a fundamental trade-off between latency and accuracy. A shorter $T$ reduces latency by shortening spike trains but risks creating an information bottleneck that degrades performance. Conversely, a longer $T$ enhances temporal resolution and accuracy at the expense of increased latency. To empirically analyze this trade-off in our CH-SNN framework, we conducted a controlled study by training sparse networks under identical settings while systematically varying $T$. The results shown in Table 15 clearly illustrate this balance and identify $T = 8$ as the optimal operating point, delivering strong performance with moderate latency.

Table 15: Performance on different timesteps.

| **Dataset** | Network | $T = 2$ | $T = 4$ | $T = 8$ | $T = 16$ |
|---|---|---|---|---|---|
| N-MNIST | 2CONV2FC | 96.20% | 98.32% | 99.15% | 99.20% |
| MNIST | 2CONV2FC | 98.55% | 98.89% | 99.53% | 99.55% |
| DVS-Gesture | 6CONV2FC | 85.60% | 90.63% | 95.45% | 95.45% |
| CIFAR-10 | 6CONV2FC | 75.33% | 87.21% | 94.60% | 94.71% |
| CIFAR-100 | 6CONV2FC | 72.04% | 73.52% | 75.22% | 75.34% |

### A.12 EXTENSION EXPERIMENT

To validate our approach on more complex datasets and deeper network architectures, we evaluate our CH-SNN against DPAP (Han et al., 2025b), currently the leading sparse training method for SNNs, on the Tiny-ImageNet and ImageNet datasets using a ResNet-18-based SNN architecture. To demonstrate the scalability of our approach, we apply the same core methodology to two distinct layer types. CH-CNN (Hanming et al., 2025) is used to sparsify the convolutional layers, while CH-SNN sparsifies the linear layers in the SNN architecture. The comparative top-1 accuracy results are summarized in the table 16, demonstrating the competitive performance and strong scalability of our method.

Table 16: Performance on Tiny ImageNet and ImageNet.

| **Dataset** | **Method** | **Sparsity** | **ACC.(sparse)** | **ACC.(dense)** | **Acc loss** |
|---|---|---|---|---|---|
| **Tiny ImageNet** | CH-SNN | 69.37% | 44.97% | 45.98% | 1.01% |
| **ImageNet** | CH-SNN | 66.94% | 62.77% | 63.62% | 0.85% |
| | DPAP | 51.71% | 60.41% | 65.74% | 5.33% |
| | DPAP | 37.76% | 63.35% | 65.74% | 2.39% |
| | DPAP | 22.69% | 63.74% | 65.74% | 2.00% |

On the Tiny-ImageNet dataset, we report the results for CH-SNN as comparative results for DPAP are not available in the literature. Our CH-SNN model achieves an accuracy of 44.97% while maintaining a 69.37% sparsity rate, a performance level that is comparable to the dense baseline.

On the ImageNet dataset, We directly report the results for DPAP and its corresponding FC baseline from the original publication. This approach ensures a faithful comparison and avoids potential implementation discrepancies. For our CH-SNN, we adopted the identical ResNet-18 architecture used in DPAP and conducted a fair comparison by training both a fully-connected (FC) model and a sparse CH-SNN model under the same experimental protocol. The results demonstrate that CH-SNN achieves a high structural sparsity of 66.94% while attaining competitive performance, with only a 0.85% accuracy drop compared to the FC baseline.

### A.13 USAGE OF LARGE LANGUAGE MODELS

In the process of preparing this manuscript, we utilized the DeepSeek large language model to assist in polishing the English writing and refining the wording of the Abstract, Introduction and Conclusion sections. The core ideas, theoretical contributions, experimental design, data analysis, and results remain entirely the work of the authors. The authors take full responsibility for the entire content of this paper.

