# OpenReview forum: "Cannistraci-Hebb Training on Ultra-Sparse Spiking Neural Networks"
_ICLR.cc/2026/Conference — ICLR 2026 Poster_

### Official Review · Reviewer_nSp9 · 2025-10-27

**Soundness:** 3
**Presentation:** 3
**Contribution:** 3
**Rating:** 6
**Confidence:** 3

**Summary:**

This paper introduces CH-SNN (Cannistraci-Hebb Spiking Neural Network), which is a four-stage dynamic sparse training framework for ultra-sparse spiking neural networks (SNNs). Extensive experiments on six datasets show CH-SNN achieves performance comparable to FC networks even at ultra-high levels of sparsity

**Strengths:**

Originality
(1) Integrates Cannistraci-Hebb theory, originally from complex network science, into SNN sparse training.
(2) Introduces two novel initialization schemes (SSCTI, SSWI) specifically designed for spike-based learning.

Quality
(1) Extensive experiments across six datasets show robustness and generalizability.
(2) Includes ablation, sensitivity, and hardware efficiency analyses, showing methodological thoroughness.

Clarity
The paper is clearly structured, with each stage of CH-SNN well explained. The biological and theoretical motivations are well linked to the computational framework.

**Weaknesses:**

(1) Insufficient analysis of temporal dynamics:
The paper emphasizes structural sparsity but offers limited insight into the temporal spike dynamics.
(2) Clarity of comparison fairness:
It is not entirely clear whether all baseline methods were reimplemented under identical experimental conditions. The paper does not specify whether these results were reproduced using a unified experimental setup or directly taken from prior publications, which affects the transparency and comparability of the reported performance gains.

**Questions:**

(1) Temporal sparsity–accuracy trade-off: Have the authors analyzed the effect of increasing temporal sparsity on latency or robustness?
(2) Reproducibility: Please clarify whether all baseline methods were reimplemented under identical training conditions or if their results were directly adopted from previous studies.

---

> ### Author Response · Authors · 2025-12-02
> **Reply to the Reviewer nSp9**
>
> Dear Reviewer nSp9,
>
> Thank you for your review of our paper. Below are the replies to your weaknesses and questions.
>
> **Reply to weakness 1**
>
> We now perform the analysis of increasing temporal sparsity on latency or robustness as for the reply to the **Question 1** below.
>
> **Reply to weakness 2**
>
> We now address the comparison fairness as for the reply to the **Question 2** below.
>
> **Reply to question 1**
>
> We appreciate the reviewer's insightful question. We fully agree that analyzing the impact of temporal sparsity on latency and robustness is crucial for the practical application of spiking neural networks.
>
> **1. Latency analysis**
>
> For spiking neural networks deployed on edge AI devices, latency is directly related to the number of synaptic operations (SOPs). We conducted experiments to analyze synaptic operations, results in the table below show that across four datasets, the sparse network trained with CH-SNN achieves an average reduction of two orders of magnitude in synaptic operations compared to the fully-connected network. Consequently, on edge devices, the CH-SNN sparse network delivers significantly lower latency while maintaining accuracy comparable to the fully-connected network.
>
> | Dataset       | Spike Count      | SOPs            | Firing Rate | Link Sparsity | Node Sparsity | Acc.     | Energy   |
> |---------------|------------------|-----------------|-------------|---------------|---------------|----------|----------|
> | **MNIST**     | 6.1 × 10⁸        | 6.3 × 10¹¹      | 31.22%      | 0%            | 0%            | 97.29%   | 948mJ    |
> |               | 3.3 × 10⁸        | 3.2 × 10¹⁰      | 16.83%      | 94.59%        | 23.47%        | 97.56%   | 48mJ     |
> | **DVS-Gesture** | 2.2 × 10⁷        | 5.2 × 10¹⁰      | 4.02%       | 0%            | 0%            | 89.02%   | 78mJ     |
> |               | 1.3 × 10⁷        | 5.0 × 10⁸       | 2.45%       | 98.84%        | 12.30%        | 91.29%   | 0.8mJ    |
> | **N-MNIST**   | 2.5 × 10⁸        | 1.4 × 10¹¹      | 10.42%      | 0%            | 0%            | 96.38%   | 216mJ   |
> |               | 1.2 × 10⁸        | 2.9 × 10⁹       | 4.72%       | 98.46%        | 41.90%        | 96.20%   | 4.4mJ    |
> | **CIFAR-10**  | 6.3 × 10⁷        | 2.8 × 10¹⁰      | 30.70%      | 0%            | 0%            | 80.67%   | 41mJ     |
> |               | 1.9 × 10⁷        | 4.8 × 10⁸       | 9.28%       | 93.78%        | 0%            | 82.84%   | 0.7mJ    |

---

> ### Author Response · Authors · 2025-12-02
> **Reply to the Reviewer nSp9**
>
> **Reply to question 1**
>
> **2. Robustness analysis**
>
> To evaluate the robustness of CH-SNN, we conduct experiments where the models, trained on clean data, are exposed to corrupted inputs during testing. The corruption involves three noise types: (1) Bit-Flip Noise. This noise randomly flips 0s to 1s and 1s to 0s. Its destructive nature stems from a dual effect: it corrupts the input by both adding spurious spikes and removing authentic ones. (2) False-Spike Noise. This corruption randomly changes 0s to 1s, which generates extraneous spikes. This directly compromises the timing precision fundamental to SNN operation. (3) Spike-Dropout Noise. This type randomly changes 1s to 0s, thereby dropping genuine spikes. The testing protocol involved corrupting the input with three noise types across a range of intensities. The noise intensity was controlled by the noise ratio, defined as the proportion of timesteps in the input spike train that are altered.
>
> The robustness of the model is quantified using the Relative Performance Degradation Rate (RPDR), which is formally defined as Equation (1)  $$\mathrm{RPDR}=\frac{ACC_{clean}-ACC_{noise}}{ACC_{clean}} \tag{1}$$
>
> where $ACC_{clean}$ is the baseline accuracy on unperturbed data, and $ACC_{noise}$ is the accuracy evaluated under a specific noise corruption. This metric is interpreted as follows: a smaller RPDR denotes stronger robustness, as it reflects a smaller relative drop in accuracy. Table below presents a comprehensive summary, detailing the RPDR for each noise type alongside the mean RPDR aggregated over all noise conditions.
>
> **Table 1: The BPDR of the model.**
> | Dataset      | Method  | Bit-flip | False-spike | Spike-dropout | Average |
> |--------------|---------|----------|-------------|---------------|---------|
> | MNIST        | FC      | 11.95%   | 7.41%       | 11.52%        | 10.29%  |
> |              | CH-SNN  | 13.75%   | 9.52%       | 13.01%        | 12.09%  |
> | N-MNIST      | FC      | 18.38%   | 17.73%      | 0.03%         | 12.05%  |
> |              | CH-SNN  | 17.64%   | 16.95%      | 0.13%         | 11.57%  |
> | CIFAR-10     | FC      | 32.95%   | 31.46%      | 0.35%         | 21.59%  |
> |              | CH-SNN  | 41.07%   | 41.27%      | 0.44%         | 27.59%  |
> | DVS-Gesture  | FC      | 14.19%   | 16.09%      | -0.22%        | 10.02%  |
> |              | CH-SNN  | 12.93%   | 12.39%      | -0.33%        | 8.33%   |
>
> Analysis on non-spiking datasets (MNIST, CIFAR-10). On these converted datasets, where static images are encoded into spike trains, the FC baseline exhibits greater robustness on average compared to the CH-SNN. We postulate that the densely and uniformly distributed information in static images aligns better with the FC network's inherent redundancy, granting it higher fault tolerance. The structural sparsity of CH-SNN, while beneficial in other contexts, leads to increased temporal sparsity that offers no robustness advantage here. Consequently, achieving robustness on such tasks would necessitate a design with lower structural sparsity.
>
> Analysis on native spiking datasets (DVS-Gesture, N-MNIST). The scenario reverses for native spiking data. When subjected to the most destructive noise types (bit-flip and false-spike), CH-SNN consistently shows a lower performance degradation rate than its FC counterpart. This result underscores that CH-SNN's sparse architecture effectively leverages the inherent sparsity of the data to filter out corrupting noise and protect crucial information, thus validating its stronger robustness for event-based computation.
> You can see more details in **Appendix A.9**.
>
> **Reply to question 2**
>
> We thank the reviewer for raising this important question. Regarding the comparison with baseline methods, we followed the principles below and would like to clarify as follows:
>
> SD-SNN: Since the code is available, we reimplemented this method and compared it with our CH-SNN and the fully-connected baseline under identical training conditions, including hyperparameters, and data preprocessing. This ensures a completely fair comparison with SD-SNN.
>
> Other methods (e.g., Grad R): As their code has not been released, we were unable to fully reimplement their training pipelines. Therefore, the experimental results for these methods, including both their reported baseline performance and sparse model performance, are directly adopted from their original papers.
>
> We confirm that both our baseline method and all sparse training approaches were evaluated under identical training configurations. These comparisons are therefore fair and reliable. The purpose of comparing CH-SNN with other sparse training methods is to demonstrate, in a broader context, its comprehensive performance in terms of both achieved sparsity and accuracy.
>
> To address the Reviewer concern, we now added a section in the **Appendix A.10** that summarizes and reports accurately these guidelines to reproduce and compare the results.

---

### Official Review · Reviewer_9DG8 · 2025-10-27

**Soundness:** 3
**Presentation:** 3
**Contribution:** 3
**Rating:** 6
**Confidence:** 4

**Summary:**

This paper presents CH-SNN, a novel four-stage dynamic sparse training framework for spiking neural networks (SNNs) that achieves ultra-high structural sparsity while maintaining/improving accuracy compared to baselines. Extensive experiments on six datasets and three architectures demonstrate consistent performance and energy advantages.

**Strengths:**

1. Strong novelty and cross-disciplinary contribution: bridges network science (Cannistraci-Hebb theory) with neuromorphic learning, introducing a biologically and topologically inspired sparse training approach.
2. Comprehensive experimental validation: covers multiple datasets, architectures.
3. Clear modular structure: the four-stage design (SSCTI, SSWI, LRS, CH3-L3) is intuitive and extensible to other SNNs, which provides a reasonable baseline for SNN training.

**Weaknesses:**

1. Theoretical insufficiency: The paper lacks formal analysis of the convergence and stability of the CH3-L3 regrowth dynamics.
2. Scalability questions: Experiments are limited to medium-scale datasets. The framework’s behaviour on larger datasets (e.g., ImageNet or DVS-CIFAR100) remains untested.
3. Biological claim ambiguity: The connection to Hebbian principles is mostly conceptual; empirical neuroscientific grounding is minimal.

**Questions:**

1. Could the authors provide a theoretical argument or empirical evidence that the CH3-L3 regrowth mechanism guarantees stability or avoids redundant regrowth loops?
2. How sensitive is CH-SNN to the hyperparameters controlling sparsity ratio, pruning frequency, and regrowth sampling distribution?
3. The timestep of SNN is 8 according to this paper, could the authors explained the performance on different timesteps?
4. Is the CH3-L3 topological regrowth biologically interpretable in terms of synaptic rewiring or STDP-like plasticity?

---

> ### Author Response · Authors · 2025-12-02
> **Reply to the Reviewer 9DG8**
>
> Dear Reviewer 9DG8,
>
> Thank you for your review of our paper. Below are the replies to your weaknesses and questions.
>
> **Reply to Weakness 1**
>
> We now address the lack of formal analysis of the convergence and stability of the CH3-L3 regrowth dynamics, as requested in response to **Question 1**.
>
> **Reply to Weakness 2**
>
> We thank the reviewer for valuable suggestion regarding the scalability of our method. We fully agree on the importance of validating our approach on more complex datasets and deeper network architectures. To address this point directly, we have conducted extensive supplementary experiments. Specifically, we evaluated our CH-SNN against DPAP **[1]**, currently the leading sparse training method for SNNs, on the Tiny-ImageNet and ImageNet datasets using a ResNet-18-based SNN architecture. The comparative top-1 accuracy results are summarized in the table below, demonstrating the competitive performance and scalability of our method.
>
> | Dataset       | Method | Sparsity | Acc. (sparse) | Acc. (FC) | Acc loss |
> |---------------|--------|----------|---------------|-----------|----------|
> | Tiny ImageNet | CH-SNN | 69.37%   | 44.97%        | 45.98%    | 1.01%    |
> | ImageNet      | CH-SNN | 66.94%   | 62.77%        | 63.62%    | 0.85%    |
> | ImageNet      | DPAP   | 51.71%   | 60.41%        | 65.74%    | 5.33%    |
> | ImageNet      | DPAP   | 37.76%   | 63.35%        | 65.74%    | 2.39%    |
> | ImageNet      | DPAP   | 22.69%   | 63.74%        | 65.74%    | 2.00%    |
>
> On the Tiny-ImageNet dataset, we report the results for CH-SNN as comparative results for DPAP are not available in the literature. Our CH-SNN model achieves an accuracy of 44.97% while maintaining a 69.37% sparsity rate, a performance level that is comparable to the fully-connected (FC) baseline.
>
> On the ImageNet dataset, due to time constraints that precluded a full re-implementation, we directly report the results for DPAP and its corresponding FC baseline from the original publication. This approach ensures a faithful comparison and avoids potential implementation discrepancies. For our CH-SNN, we adopted the identical ResNet-18 architecture used in DPAP and conducted a fair comparison by training both a fully-connected model and a sparse model with CH-SNN under the same experimental protocol. The results demonstrate that CH-SNN achieves a high structural sparsity of 66.94% while attaining competitive performance, with only a 0.85% accuracy drop compared to the dense FC baseline.
>
> We believe these supplementary experiments adequately address the concern regarding scalability and highlight the potential of CH-SNN for large-scale, practical applications.
>
> **References**
>
> [1] B. Han, F. Zhao, Y. Zeng and G. Shen, "Developmental Plasticity-Inspired Adaptive Pruning for Deep Spiking and Artificial Neural Networks," in IEEE Transactions on Pattern Analysis and Machine Intelligence, vol. 47, no. 1, pp. 240-251, Jan. 2025, doi: 10.1109/TPAMI.2024.3467268.

---

> ### Author Response · Authors · 2025-12-02
> **Reply to the Reviewer 9DG8**
>
> **Reply to weakness 3**
>
> Thank you for raising this important point regarding the biological ambiguity in our claims. We appreciate the opportunity to clarify and strengthen our response. While we agree that the direct implementation in our CH-SNN framework is conceptual—serving as an abstraction for computational efficiency—the Cannistraci-Hebb (CH) theory itself has robust empirical neuroscientific grounding **[1, 2, 3, 4]**, extending classical Hebbian principles ("cells that fire together wire together") to higher-order network topologies in brain connectomes.
>
> Specifically, CH theory reinterprets Hebbian learning through an "epitopological" lens, to any complex system. This is validated in brain networks and many other types of complex networks **[1, 3]**, where CH topological link-prediction methods were applied to real complex networks. In particular, on brain connectome data: In-vivo mouse primary visual cortex connectomes (from two-photon calcium imaging and electron microscopy), in-silico macaque cortical networks, and C. elegans frontal ganglia networks. Synapses were randomly deleted (10-50%), and re-predicted using CH-inspired indices.
>
> CH-based methods outperformed classical indices (e.g., Common Neighbors) by up to 186% in prediction precision (measured by Area Under the Prediction Power Curve, AUPPC, with p<0.05 statistical significance over 1000 trials). This demonstrates that new synaptic links preferentially form within local communities, aligning with neuroplasticity observations where persistent synapse sets remodel during learning.
>
> CH extends Hebb's 1949 hypothesis by modeling how co-activated neurons' synapses form "local communities," predicting growth based on topological features like common neighbors, and internal links versus external links balance. This has implications for disorders like autism (Intense World Syndrome), involving hyper-plasticity in local circuits.
>
> In neuromorphic computing, many bio-inspired methods (e.g., STDP in SNNs) are similarly conceptual abstractions without exact biological replication, yet they advance energy-efficient AI. Our work follows this tradition, using CH to achieve ultra-sparsity with performance gains.
>
> **References**
>
> [1] Cannistraci, C.V. Modelling Self-Organization in Complex Networks Via a Brain-Inspired Network Automata Theory Improves Link Reliability in Protein Interactomes. Sci Rep 8, 15760 (2018). https://doi.org/10.1038/s41598-018-33576-8
>
> [2] Zhang, Y.; Zhao, J.; Wu, W.; Muscoloni, A.; Cannistraci, C. V. Epitopological Learning and Cannistraci-Hebb Network Shape Intelligence Brain-Inspired Theory for Ultra-Sparse Advantage in Deep Learning. The Twelfth International Conference on Learning Representations (ICLR) 2024.
>
> [3] J Zhao, A Muscoloni, U Michieli, Y Zhang, CV Cannistraci. Adaptive Cannistraci-Hebb Network Automata Modelling of Complex Networks for Path-based Link Prediction. Thirty-Ninth Conference on Neural Information Processing Systems (NeurIPS 2025).
>
> [4] Zhang, Y., Cerretti, D., Zhao, J., Wu, W., Liao, Z., Michieli, U., & Cannistraci, C.V. (2025). Brain network science modelling of sparse neural networks enables Transformers and LLMs to perform as fully connected. Thirty-Ninth Conference on Neural Information Processing Systems (NeurIPS 2025).

---

> ### Author Response · Authors · 2025-12-02
> **Reply to the Reviewer 9DG8**
>
> **Reply to question 1**
>
> We sincerely thank the reviewer for raising this crucial point regarding the stability and avoidance of redundant regrowth loops in the CH3-L3 mechanism. In our work, we address this issue through a combination of theoretical design of the CH3-L3 rule itself and an explicit empirical early-stop strategy, which together ensure training stability and prevent futile regrowth cycles.
>
> The CH3-L3 regrowth score is derived from the Cannistraci-Hebb theory of local-community organization **[1, 2, 3]**. The core of its stability lies in Equation (1):
> $$CH3-L3\left(u,v\right)=\sum_{z_1,z_2\in l3\left(u,v\right)}\frac{1}{\sqrt{\left(1+de_{z_1}\right)\times\left(1+de_{z_2}\right)}} \tag{1}$$
> The term $d_ez$ (the external degree of a common neighbor $z$) acts as a penalty term. It inherently discourages the formation of new links that would connect to nodes that are already highly connected outside their immediate local community. You can see more details in **Appendix A.2**. This design naturally prevents the mechanism from getting stuck in redundant loops of adding and removing the same connections. It preferentially strengthens connections within structurally isolated communities, which is a stable, self-reinforcing process. It is a gradient-free, topology-driven predictive network automata **[3, 4]**   that guides the network towards a sparse, hub-and-community structure **[2]**.
>
> While CH3-L3 is inherently stable, we proactively introduced an early-stop mechanism in the CH-SNN framework to monitor and halt any potential redundancy. We track the overlap rate between the set of links removed and the set of links regrown in each topological update cycle. Once this overlap rate exceeds a high threshold (e.g., 95%), it signals that the network topology has stabilized. At this point, we permanently stop the topological evolution for that layer and focus solely on weight update.
>
> Furthermore, our framework includes a chain removal, as shown in **section 3.2.3**, that permanently removes inactive neurons. Since these neurons cannot attract new connections via CH3-L3, their removal prevents structural dead-ends and further contributes to overall training stability and efficiency.
>
> Our extensive experiments across multiple datasets and architectures, where CH-SNN consistently converges and outperforms baselines, serve as empirical validation of this stable behavior.
>
> Finally, the previous studies on dynamic sparse training (DST) never raised a concern of stable convergence or meaningful structure learning because the DST methodology selects which link should be removed during the training by using the weight update, this ensures a convergence of the model towards meaningful structures **[2, 6, 7]** even with a random predictor such as SET **[5]**.
>
> We thank the reviewer again for this question and hope this clarification adequately addresses their concern. To address the Reviewer concern, we now added a section in the **Appendix A.2.1** that summarizes and reports accurately this discussion about how CH3-L3 regrowth mechanism guarantees stability or avoids redundant regrowth loops.
>
> **References**
>
> [1] Cannistraci, C.V. Modelling Self-Organization in Complex Networks Via a Brain-Inspired Network Automata Theory Improves Link Reliability in Protein Interactomes. Sci Rep 8, 15760 (2018). https://doi.org/10.1038/s41598-018-33576-8
>
> [2] Zhang, Y.; Zhao, J.; Wu, W.; Muscoloni, A.; Cannistraci, C. V. Epitopological Learning and Cannistraci-Hebb Network Shape Intelligence Brain-Inspired Theory for Ultra-Sparse Advantage in Deep Learning. The Twelfth International Conference on Learning Representations (ICLR) 2024.
>
> [3] J Zhao, A Muscoloni, U Michieli, Y Zhang, CV Cannistraci. Adaptive Cannistraci-Hebb Network Automata Modelling of Complex Networks for Path-based Link Prediction. Thirty-Ninth Conference on Neural Information Processing Systems (NeurIPS 2025).
>
> [4] Zhang, Y., Cerretti, D., Zhao, J., Wu, W., Liao, Z., Michieli, U., & Cannistraci, C.V. (2025). Brain network science modelling of sparse neural networks enables Transformers and LLMs to perform as fully connected. Thirty-Ninth Conference on Neural Information Processing Systems (NeurIPS 2025).
>
> [5] Mocanu, D.C., Mocanu, E., Stone, P. et al. Scalable training of artificial neural networks with adaptive sparse connectivity inspired by network science. Nat Commun 9, 2383 (2018). https://doi.org/10.1038/s41467-018-04316-3
>
> [6] Utku Evci, Trevor Gale, Jacob Menick, Pablo Samuel Castro, and Erich Elsen. 2020. Rigging the lottery: making all tickets winners. In Proceedings of the 37th International Conference on Machine Learning (ICML'20), Vol. 119. JMLR.org, Article 276, 2943–2952.
>
> [7] Zhang, Y., Zhao, J., Liao, Z., Wu, W., Michieli, U., & Cannistraci, C. V. (2024). Brain-Inspired Sparse Training in MLP and Transformers with Network Science Modeling via Cannistraci-Hebb Soft Rule. Preprints. https://doi.org/10.20944/preprints202406.1136.v1

---

> ### Author Response · Authors · 2025-12-02
> **Reply to the Reviewer 9DG8**
>
> **Reply to question 2**
>
> **1. Sparsity ratio**
>
> We conducted tests under varying levels of structural sparsity (70%, 80%, 95%, and 99%). The results show that CH-SNN achieves performance comparable to the fully-connected network across all these sparsity levels. While there is a slight downward trend in accuracy as sparsity increases, the magnitude of degradation remains minimal. This demonstrates that CH-SNN maintains performance on par with the fully-connected network even at ultra sparsity levels ($\geq99\\%$). The results are presented in the table below.
>
> | Dataset      | FC     | 70% Sparsity | 80% Sparsity | 95% Sparsity | 99% Sparsity |
> |--------------|--------|--------------|--------------|--------------|--------------|
> | N-MNIST      | 97.16% | 97.22%       | 97.29%       | 97.21%       | 96.04%       |
> | MNIST        | 98.09% | 98.23%       | 98.11%       | 97.56%       | 97.47%       |
> | DVS-Gesture  | 89.02% | 89.02%       | 88.64%       | 91.29%       | 89.77%       |
> | CIFAR-10     | 79.58% | 81.42%       | 81.73%       | 82.84%       | 81.13%       |
>
> **2. Pruning frequency**
>
> Dynamic Pruning Ratio ($\zeta$): To evaluate the impact of different pruning rate strategies on model performance, we first test the dynamic pruning rate strategy as defined in Equation (1).
> $$\zeta=\frac{\zeta}{2}\times\left(1+\mathrm{cos}\left(\frac{\pi\times e p o c h}{total\ \ pochs}\right)\right) \tag{1}$$
>
> We adjust various decay starting points (0.5, 0.4, 0.3, 0.2, 0.1), and the experimental results are shown in table as below. It can be observed that CH-SNN exhibits negligible performance variation across different initial pruning rates, demonstrating strong stability in response to changes in the pruning rate. The model consistently achieves strong performance under the predefined sparsity targets.
>
> | Dataset      | $\zeta=0.5$  | $\zeta=0.4$  | $\zeta=0.3$  | $\zeta=0.2$  | $\zeta=0.1$  |
> |--------------|--------|--------|--------|--------|--------|
> | N-MNIST      | 97.14% | 96.91% | 96.96% | 97.04% | 97.10% |
> | DVS-Gesture  | 89.39% | 90.53% | 89.02% | 88.64% | 89.39% |
> | MNIST        | 97.98% | 97.98% | 97.98% | 97.98% | 97.98% |
> | CIFAR-10     | 82.64% | 82.24% | 82.84% | 83.03% | 82.75% |
>
> Static Pruning Ratio: Similarly, we evaluate a static pruning rate strategy. Unlike the dynamic approach, the static pruning rate remains constant at its initial value throughout training. We test multiple starting values for the static pruning rate, and the results are presented in table below. CH-SNN shows minimal performance variation across different static pruning rates. It is worth noting that compared to the dynamic pruning strategy, the static approach generally leads to a slight decrease in overall accuracy.
>
> | Dataset      | $\zeta=0.5$  | $\zeta=0.4$  | $\zeta=0.3$  | $\zeta=0.2$  | $\zeta=0.1$  |
> |--------------|--------|--------|--------|--------|--------|
> | N-MNIST | 96.90% | 96.87% | 96.84% | 96.81% | 97.10% |
> | DVS-Gesture  | 88.64% | 89.39% | 89.02% | 89.02% | 88.64% |
> | MNIST    | 97.98% | 97.98% | 97.98% | 97.98% | 97.98% |
> | CIFAR-10  | 82.24% | 82.26% | 82.53% | 82.63% | 82.25% |
>
> You can see more sensitivity test result in the **Appendix A.7**.
>
> **3. Regrowth sampling distribution**
>
> During link regrowth, we compute the link regrowth score ($CH3-L3\left(u,v\right)$) for nonexistent links, as formulated in Equation (2).
> $$ CH3-L3\left(u,v\right)={\sum_{z_1,z_2\in l3\left(u,v\right)}\frac{1}{\sqrt{\left(1+de_{z_1}\right)\times\left(1+de_{z_2}\right)}}}^\frac{\delta}{1-\delta} \tag{2}$$
>
> The parameter $\delta$ controls the sampling distribution. When $\delta\ =\ 0$, it means the $CH3-L3\left(u,v\right)$ is identical for all links. In this scenario, the corresponding sampling method is random sampling, where links are randomly selected and regrown. When $\delta=\ 1$, the sampling becomes deterministic, and links are regrown directly based on their $CH3-L3\left(u,v\right)$ values. When $\delta=\ 0.5$, it means sampling from a multinomial distribution based on the $CH3-L3\left(u,v\right)$ values. We tested three different sampling strategies ($\delta$=0, 0.5, and 1). The experimental results presented in the table below demonstrate that the highest accuracy is achieved when $\delta=\ 0.5$. Therefore, in all experiments reported in this paper, we consistently set $\delta=\ 0.5$. You can see more details in **Appendix A.3**.
>
> | Dataset      | $\delta=1$    | $\delta=0.5$  | $\delta=0$ |
> |--------------|--------|--------|--------|
> | N-MNIST      | 96.30% | 97.21% | 96.63% |
> | MNIST        | 98.28% | 98.40% | 98.30% |
> | DVS-GESTURE  | 89.02% | 91.29% | 90.53% |
> | CIFAR-10     | 78.87% | 82.84% | 77.68% |

---

> ### Author Response · Authors · 2025-12-02
> **Reply to the Reviewer 9DG8**
>
> **Reply to question 3**
>
> The choice of timesteps ($T$) is critical in SNNs as it creates a direct trade-off between latency and accuracy. Fewer timesteps result in shorter spike trains, which reduces latency but imposes an information bottleneck that can degrade model performance. Increasing the timesteps enhances the network's temporal resolution and information capacity, typically improving accuracy at the cost of increased latency. To quantitatively analyze this balance for our CH-SNN model, we performed a controlled study, training sparse networks under identical settings while systematically varying $T$. The experimental results, presented in the following table, clearly demonstrate this trade-off.
>
> | Dataset      | Network      | $T=2$     | $T=4$     | $T=8$     | $T=16$    |
> |--------------|--------------|---------|---------|---------|---------|
> | N-MNIST      | 2Conv 2FC    | 96.20%  | 98.32%  | 99.15%  | 99.20%  |
> | MNIST        | 2Conv 2FC    | 98.55%  | 98.89%  | 99.53%  | 99.55%  |
> | DVS-Gesture  | 6Conv 2FC    | 85.60%  | 90.63%  | 95.45%  | 95.45%  |
> | CIFAR-10     | 6Conv 2FC    | 75.33%  | 87.21%  | 94.60%  | 94.71%  |
> | CIFAR-100    | 6Conv 2FC    | 72.04%  | 73.52%  | 75.22%  | 75.34%  |
>
> The experimental results identify 8 timesteps as the optimal setting, achieving a favorable balance between model performance and computational latency. Consequently, we adopt $T=8$ for all experiments reported in this paper.
>
> **Reply to question 4**
>
> Yes, the CH3-L3 topological regrowth mechanism is biologically interpretable, as it captures the functional principles of synaptic turnover and rewiring and STDP-like plasticity at a network-structural level.
>
> Although CH3-L3 does not simulate precise spike timing, it operationalizes the Hebbian idea that "neurons that fire together, wire together." It identifies neuron pairs that are structurally correlated, i.e., share common neighbors and belong to the same local community, and predicts new links between them. This process mimics structural synaptic rewiring in the brain, where connections are dynamically formed based on functional relatedness.
>
> Moreover, the local community structure emphasized by CH3-L3 can be viewed as a topological outcome of fine-grained STDP-like processes. Over time, neurons that exhibit correlated spiking activity tend to form tightly interconnected clusters. CH3-L3 leverages this structural signature to guide link regrowth, effectively performing a form of topology-driven, STDP-inspired plasticity.
> In summary, while not a direct model of biological STDP, CH3-L3 provides a functionally analogous and biologically plausible mechanism for synaptic rewiring in dynamic sparse networks.
>
> To address the Reviewer concern, we now added these new texts in the **Appendix A.2.2** that summarizes and reports accurately this discussion reported above.

---

### Official Review · Reviewer_zWas · 2025-10-30

**Soundness:** 3
**Presentation:** 3
**Contribution:** 3
**Rating:** 6
**Confidence:** 2

**Summary:**

This paper introduces Cannistraci-Hebb Spiking Neural Network (CH-SNN), a dynamic sparse training framework for ultra-sparse SNNs. Extensive experiments are conducted on six datasets, demonstrating that CH-SNN achieves high sparsity while retaining or improving accuracy over fully connected baselines.

**Strengths:**

The paper presents an approach to ultra-sparse SNN training, integrating initialization, pruning methods, and Cannistraci-Hebb-inspired topological regrowth.
Experiments were conducted on multiple datasets, and thorough ablation experiments and sensitivity analyses were carried out.
The framework is implemented on a hardware-friendly algorithm S-TP, achieving significant gains in energy efficiency.

**Weaknesses:**

Some key areas of the mathematical description, particularly around pruning and regrowth, lack sufficient clarity.

**Questions:**

Can the authors clarify the multinomial sampling procedure in the pruning step, how exactly the link removal score (LRS) is converted into actual pruning decisions?
How accurate is the SSWI initialization method under varying degrees of input temporal sparsity, structural connection sparsity,and spike threshod？

---

> ### Author Response · Authors · 2025-12-02
> **Reply to the Reviewer zWas**
>
> Dear Reviewer zWas,
>
> Thank you for your review of our paper. Below are the replies to your weakness and question.
>
> **Reply to weakness**
>
> We sincerely thank the reviewer for this valuable feedback. As requested, we have now provided a detailed explanation of both the pruning and regrowth processes, along with the complete mathematical formulations. All related equations have been included in **Appendix A.3**. For your convenience, you may also refer directly to our response to **Question**.
>
> **Reply to question**
>
> **1. Network pruning and regrowth**
>
> During dynamic sparse training, we generate a sparse connectivity matrix $C$, where $C_{ij}=1$ indicates the presence of a link between nodes $i$ and $j$, and $C_{ij}=0$ indicates no link between the nodes. After each training epoch, a process of network pruning and network regrowth is performed. Correspondingly, the sparse connectivity matrix $C$ is updated.
>
> During the link removal phase, we calculate the Link Removal Score ($LRS$) for the existing links (where $C_{ij}=1$). The $LRS_{ij}$ is computed as shown in Equation (1).  $$ LRS_{ij}=\left(\frac{\left|W_{ij}\right|}{1+\sum_{i}\left|W_{ij}\right|}+\frac{\left|W_{ij}\right|}{1+\sum_{j}\left|W_{ij}\right|}\right)^\frac{\delta}{1-\delta} \tag{1} $$
>
> The parameter $\delta$ controls the sampling distribution. When $\delta=\ 0$, it means the $LRS$ is identical for all links. In this scenario, the corresponding sampling method is random sampling, where links are randomly selected and removed. When $\delta=\ 1$, the sampling becomes deterministic, and links are removed directly based on their $LRS$ values. When $\delta=\ 0.5$, it means sampling from a multinomial distribution based on the $LRS$ values. In this situation, we calculate the link removal probability $p_{ij}^{\left(removal\right)}$ using the $LRS_{ij}$, as shown in Equation (2).  $$p_{ij}^{\left(removal\right)}=\frac{LRS_{ij}}{\sum_{ij}{LRS_{ij}}} \tag{2}$$
>
> Subsequently, based on the link removal probabilities, we remove a certain proportion ($\zeta$) of links as specified in Equation (3), completing the link removal process. $$
> C_{ij} =
> \begin{cases}
> 1 & 1 - p_{ij}^{(\text{removal})}  \\\\
> 0 & p_{ij}^{(\text{removal})}
> \end{cases}
> \quad \quad
> i, j \in  \\{ a, b \mid C_{ab} = 1 \\}
>   \tag{3}$$
> During link regrowth, we compute the link regrowth score ($CH3-L3\left(u,v\right)$) for nonexistent links (where $C_{uv}=0$), as formulated in Equation (4). $$ CH3-L3\left(u,v\right)={\sum_{z_1,z_2\in l3\left(u,v\right)}\frac{1}{\sqrt{\left(1+de_{z_1}\right)\times\left(1+de_{z_2}\right)}}}^\frac{\delta}{1-\delta} \tag{4} $$
>
> Consistent with the link removal, the parameter $\delta$ controls the sampling distribution. $\delta=0$ represents random sampling, $\delta=\ 1$ indicates that links will be directly regrown based on the $CH3-L3\left(u,v\right)$, and $\delta=\ 0.5$ signifies sampling from a multinomial distribution based on the $CH3-L3\left(u,v\right)$. Similarly, when $\delta=\ 0.5$, we compute the link regrowth probability $p_{uv}^{\left(regrowth\right)}$, as shown in Equation (5).
> $$    p_{uv}^{\left(regrowth\right)}=\frac{CH3-L3(u,v)}{\sum_{uv}{CH3-L3(u,v)}} \tag{5}$$
> Following the regrowth probabilities, the regrowth of links can be completed as shown in Equation (6). The number of regrowth links remains consistent with the number of pruned links, thereby maintaining the pre-defined overall network sparsity. $$ C_{uv} =
> \begin{cases}
> 1 & p_{uv}^{(\text{regrowth})} \\\\
> 0 & 1 - p_{uv}^{(\text{regrowth})}
> \end{cases}
> \quad \quad
> u, v \in \\{ a, b \mid C_{ab} = 0 \\} \tag{6}$$
> You can see more details in **Appendix A.3**.

---

> ### Author Response · Authors · 2025-12-02
> **Reply to the Reviewer zWas**
>
> **Reply to question**
>
> **2. Sensitivity test**
>
> **(1) Temporal sparsity**
>
> For event-based datasets such as N-MNIST and DVS-Gesture, the temporal sparsity of input features is inherently fixed. For static image datasets (e.g., MNIST, CIFAR-10), we employ rate encoding, where the temporal sparsity of the input sequence is determined by the pixel values of the images. When the pixel values remain constant, the temporal sparsity also remains fixed. To demonstrate the performance under varying sparsity levels, we present the temporal sparsity and corresponding performance metrics for each dataset in the table below:
>
> | Dataset      | Temporal Sparsity | Acc. (FC) | Acc. (95% Sparsity) |
> |--------------|-------------------|-----------|---------------------|
> | N-MNIST      | 97.83%            | 97.16%    | 97.21%              |
> | MNIST        | 86.93%            | 98.09%    | 97.56%              |
> | DVS-Gesture  | 97.34%            | 89.02%    | 91.29%              |
> | CIFAR-10     | 98.05%            | 79.58%    | 82.84%              |
>
> **(2) Structural connection sparsity**
>
> We conducted tests under varying levels of structural sparsity (70%, 80%, 95%, and 99%). The results show that CH-SNN achieves performance comparable to the fully-connected network across all these sparsity levels. While there is a slight downward trend in accuracy as sparsity increases, the magnitude of degradation remains minimal. This demonstrates that CH-SNN maintains performance on par with the fully-connected network even at ultra sparsity levels ($\geq99\\%$). The results are presented in the table below.
>
> | Dataset      | FC     | 70% Sparsity | 80% Sparsity | 95% Sparsity | 99% Sparsity |
> |--------------|--------|--------------|--------------|--------------|--------------|
> | N-MNIST      | 97.16% | 97.22%       | 97.29%       | 97.21%       | 96.04%       |
> | MNIST        | 98.09% | 98.23%       | 98.11%       | 97.56%       | 97.47%       |
> | DVS-Gesture  | 89.02% | 89.02%       | 88.64%       | 91.29%       | 89.77%       |
> | CIFAR-10     | 79.58% | 81.42%       | 81.73%       | 82.84%       | 81.13%       |
>
> **(3) Spike threshold**
>
> For the fully connected network, we employ Kaiming initialization **[1]**, while for the sparse network, we use Sparse Spike Weight Initialization (SSWI). Since SSWI incorporates neuronal threshold ($\theta$) information into the initialization process, the sparse network trained with CH-SNN exhibits low sensitivity to variations in the spike threshold hyperparameter. In contrast, Kaiming initialization does not incorporate any threshold-related information specific to spiking neural networks. As a result, the accuracy of the fully connected network is more sensitive to threshold changes and demonstrates significantly greater fluctuations. The results are presented in tables below.
>
> **Table 1: The performance of CH-SNN with 99% sparsity.**
>
> | Dataset      | Threshold-1 | Threshold-2 | Threshold-3 | Threshold-4 | Threshold-5 |
> |--------------|-------------|-------------|-------------|-------------|-------------|
> | N-MNIST      | 96.48%      | 96.54%      | 96.20%      | 96.04%      | 95.82%      |
> | MNIST        | 96.88%      | 97.44%      | 97.46%      | 97.47%      | 97.31%      |
> | DVS-Gesture  | 88.64%      | 91.29%      | 88.64%      | 89.77%      | 89.39%      |
> | CIFAR-10     | 83.41%      | 81.09%      | 80.96%      | 81.13%      | 79.54%      |
>
> **Table 2: The performance of Fully-connected network.**
>
> | Dataset      | Threshold-1 | Threshold-2 | Threshold-3 | Threshold-4 | Threshold-5 |
> |--------------|-------------|-------------|-------------|-------------|-------------|
> | N-MNIST      | 85.07%      | 88.27%      | 96.38%      | 97.16%      | 97.10%      |
> | MNIST        | 97.29%      | 97.99%      | 97.83%      | 98.09%      | 98.05%      |
> | DVS-Gesture  | 72.73%      | 75.76%      | 78.03%      | 89.02%      | 87.88%      |
> | CIFAR-10     | 74.63%      | 79.23%      | 79.13%      | 79.58%      | 80.52%      |
>
> **References**
>
> [1] He, K., Zhang, X., Ren, S., & Sun, J. (2015). Delving Deep into Rectifiers: Surpassing Human-Level Performance on ImageNet Classification. 2015 IEEE International Conference on Computer Vision (ICCV), 1026-1034.

---

### Official Review · Reviewer_iDtv · 2025-11-01

**Soundness:** 2
**Presentation:** 3
**Contribution:** 2
**Rating:** 4
**Confidence:** 3

**Summary:**

This paper proposes CH-SNN, a dynamic sparse training framework for SNNs. The method sparsifies all linear layers in SNNs through four stages: (1) sparse topology initialization (SSCTI), (2) sparse weight initialization (SSWI), (3) hybrid pruning based on link removal scores (LRS), and (4) link regrowth using CH3-L3. Experiments shows that CH-SNN achieves ultra-high structural sparsity while maintaining accuracy. The authors also report significant energy savings when deploying CH-SNN on hardware-friendly algorithm S-TP.

**Strengths:**

1. CH-SNN achieves ultra-high sparsity (>90% on some datasets) without performance degradation.

2. The four-stage framework is well-structured and includes ablation studies showing the necessity of SSCTI and SSWI for stable training under extreme sparsity.

**Weaknesses:**

1. All experiments are conducted on relatively simple tasks using very shallow networks. The lack of evaluation on more complex datasets and deeper SNNs raises serious doubts about scalability. For example, the extremely high sparsity achieved on MNIST is likely attributable to the simplicity of the task, whereas the sparsity drops significantly on CIFAR. It can be inferred that on more challenging benchmarks like ImageNet, the claimed “ultra-sparse” may not be achievable. In contrast, works like SRigL (cited in Section 2.1) have been validated on ResNet-scale models. If the authors can provide relevant evidence, I am willing to increase my rating accordingly.

2. The Spikformer results lack meaningful comparison. First, no other sparse training method is evaluated on Spikformer, making any performance comparison meaningless. Second, Transformers are typically designed for large-scale datasets, applying them to MNIST-level tasks offers little insight. Moreover, the paper never specifies the depth or width of the Spikformer used, making it impossible to assess the result’s significance.

3. The paper repeatedly highlights marginal gains (e.g., outperforms FC network by 0.16% on MNIST) as evidence of superiority. However, MNIST is nearly saturated (~99% accuracy), and such a gain is statistically negligible. This risks overstating the method’s effectiveness.

**Questions:**

1. Section 3.2.1 states that for intermediate layers (e.g., after conv or attention), SSCTI is inapplicable and replaced by uniform random initialization. Does this mean the core topological initialization method is effectively limited to the first layer? How can CH-SNN ensure stable convergence or meaningful structure learning in deeper networks where feature correlations are nontrivial?

2. On CIFAR-100, sparse models report large improvements over the baseline (Table 1). Can the authors explain why there are such large improvements? Intuitively, such a large gap strongly suggests the baseline FC models may not have been sufficiently tuned.

3. Table 1 shows Grad R achieves 91.95% accuracy on DVS-Gesture with an accuracy improvement of +7.83%. However, CH-SNN reports 95.45% accuracy with only +0.38% improvement. Is there a reporting error in the accuracy or the improvement values?

---

> ### Author Response · Authors · 2025-12-02
> **Reply to the Reviewer iDtv**
>
> Dear Reviewer iDtv,
>
> Thank you for your review of our paper. Below are the replies to your weaknesses and questions.
>
> **Reply to weakness 1**
>
>  We thank the reviewer for valuable suggestion regarding the scalability of our method. We fully agree on the importance of validating our approach on more complex datasets and deeper network architectures. To address this point directly, we have conducted extensive supplementary experiments. Specifically, we evaluated our CH-SNN against DPAP **[1]**, currently the leading sparse training method for SNNs, on the Tiny-ImageNet and ImageNet datasets using a ResNet-18-based SNN architecture. The comparative top-1 accuracy results are summarized in the table below, demonstrating the competitive performance and scalability of our method.
>
> | Dataset       | Method | Sparsity | Acc. (sparse) | Acc. (FC) | Acc loss |
> |---------------|--------|----------|---------------|-----------|----------|
> | Tiny ImageNet | CH-SNN | 69.37%   | 44.97%        | 45.98%    | 1.01%    |
> | ImageNet      | CH-SNN | 66.94%   | 62.77%        | 63.62%    | 0.85%    |
> | ImageNet      | DPAP   | 51.71%   | 60.41%        | 65.74%    | 5.33%    |
> | ImageNet      | DPAP   | 37.76%   | 63.35%        | 65.74%    | 2.39%    |
> | ImageNet      | DPAP   | 22.69%   | 63.74%        | 65.74%    | 2.00%    |
>
> On the Tiny-ImageNet dataset, we report the results for CH-SNN as comparative results for DPAP are not available in the literature. Our CH-SNN model achieves an accuracy of 44.97% while maintaining a 69.37% sparsity rate, a performance level that is comparable to the fully-connected (FC) baseline.
>
> On the ImageNet dataset, due to time constraints that precluded a full re-implementation, we directly report the results for DPAP and its corresponding FC baseline from the original publication. This approach ensures a faithful comparison and avoids potential implementation discrepancies. For our CH-SNN, we adopted the identical ResNet-18 architecture used in DPAP and conducted a fair comparison by training both a fully-connected model and a sparse model with CH-SNN under the same experimental protocol. The results demonstrate that CH-SNN achieves a high structural sparsity of 66.94% while attaining competitive performance, with only a 0.85% accuracy drop compared to the dense FC baseline.
>
> We believe these supplementary experiments adequately address the concern regarding scalability and highlight the potential of CH-SNN for large-scale, practical applications.
>
> **Reply to weakness 2**
>
> We thank the reviewer for their insightful comments regarding comparative evaluation and scalability validation. We fully agree on the importance of these aspects.
>
> Regarding comparisons with other sparse methods, we would like to emphasize that the main challenge in sparse training is to maintain competitive performance while significantly reducing network connectivity. From this perspective, the results achieved by our CH-SNN on the Spikformer architecture are notably meaningful. We did not compare only on MNIST-level tasks. For instance, on CIFAR-100 (which is notoriously more complex task than MNIST-level), CH-SNN attains 76.23% accuracy with 82.11% sparsity, which represents a +0.75% accuracy improvement over the dense baseline while drastically reducing connectivity.
>
> We have initiated large-scale experiments on the ImageNet dataset. The training process is currently underway, and we are committed to providing the complete results immediately upon their completion.
>
> For all experiments, we adopt the standard Spikformer architecture **[2]**, configured with 8 encoder layers, a hidden dimension of 512, 8 attention heads, and a timestep of 4. We added these details in the **Appendix A.5**.
>
> **References**
>
> [1] B. Han, F. Zhao, Y. Zeng and G. Shen, "Developmental Plasticity-Inspired Adaptive Pruning for Deep Spiking and Artificial Neural Networks," in IEEE Transactions on Pattern Analysis and Machine Intelligence, vol. 47, no. 1, pp. 240-251, Jan. 2025, doi: 10.1109/TPAMI.2024.3467268.
>
> [2] Zhaokun Zhou, Yuesheng Zhu, Chao He, Yaowei Wang, Shuicheng Yan, Yonghong Tian, and Li Yuan. Spikformer: When spiking neural network meets transformer, 2022. URL https: //arxiv.org/abs/2209.15425.

---

> ### Author Response · Authors · 2025-12-02
> **Reply to the Reviewer iDtv**
>
> **Reply to weakness 3**
>
> We thank the reviewer for raising the important point regarding the significance of performance improvements. We fully agree that isolated absolute accuracy gains on nearly saturated datasets like MNIST should be interpreted with caution.
>
> We would like to clarify the key intent behind reporting these results: the critical context for the 0.16% improvement on MNIST was achieved under an extreme structural sparsity of 97.75%. In machine learning, removing over 97% of network connections typically leads to catastrophic performance collapse. Therefore, maintaining performance comparable to the fully-connected baseline at any sparsity level is highly challenging. The fact that CH-SNN achieves a slight improvement, even a marginal one, under such conditions demonstrates the exceptional efficiency of the sparse topology it generates. This signifies a substantive breakthrough in addressing the core challenge of sparse training.
>
> To prevent any potential misunderstanding, we emphasize that the advantages of CH-SNN have been validated across multiple datasets, with particularly prominent results on non-saturated tasks. For instance, on the CIFAR-10 dataset, CH-SNN again achieves a 74.62% sparsity with only a minimal accuracy drop of 0.14%. On the CIFAR-100 dataset, CH-SNN attains a 74.45% sparsity while improving accuracy by 3.16%. These collective findings indicate that the performance gains achieved by CH-SNN are consistent and generalizable, rather than being specific to a single saturated dataset.
>
> **Reply to question 1**
>
> We thank the Reviewer for giving us the opportunity to clarify this key point. SSCTI is used to initialize any topology which takes the feature signal information. Therefore, this is not limited to the first layer only. Please see the newly added **Appendix A.4**. As you can see from the text, we clarify that in case of the intermediate layers in Transformer **[1]** or Spikformer **[2]**, since they do not receive any feature information, they are initialized using a network science topological initialization model. The baseline to initialize the topology of dynamic sparse training (DST) is the ER model which samples uniformly at random to the links with the given sparsity. When we write uniform random initialization, we mean that we use the ER model. There are more advanced models that can be used, for instance, the recently proposed Bipartite Receptive Field model **[3]**   and we now also show the results with this. The previous studies on DST never raised a concern of stable convergence or meaningful structure learning associated with the initialization because the DST methodology selects which link should be removed during the training by using the weight update, this ensures a convergence of the model towards meaningful structures **[4, 5, 6, 7]**.
>
> **References**
>
> [1] Ashish Vaswani, Noam Shazeer, Niki Parmar, Jakob Uszkoreit, Llion Jones, Aidan N. Gomez, Łukasz Kaiser, and Illia Polosukhin. 2017. Attention is all you need. In Proceedings of the 31st International Conference on Neural Information Processing Systems (NIPS'17). Curran Associates Inc., Red Hook, NY, USA, 6000–6010.
>
> [2] Zhou, Z., Zhu, Y., He, C., Wang, Y., Yan, S., Tian, Y., & Yuan, L. (2022). Spikformer: When Spiking Neural Network Meets Transformer. ArXiv, abs/2209.15425.
>
> [3] Zhang, Y., Cerretti, D., Zhao, J., Wu, W., Liao, Z., Michieli, U., & Cannistraci, C.V. (2025). Brain network science modelling of sparse neural networks enables Transformers and LLMs to perform as fully connected. Thirty-Ninth Conference on Neural Information Processing Systems (NeurIPS 2025).
>
> [4] Mocanu, D.C., Mocanu, E., Stone, P. et al. Scalable training of artificial neural networks with adaptive sparse connectivity inspired by network science. Nat Commun 9, 2383 (2018). https://doi.org/10.1038/s41467-018-04316-3
>
> [5] Utku Evci, Trevor Gale, Jacob Menick, Pablo Samuel Castro, and Erich Elsen. 2020. Rigging the lottery: making all tickets winners. In Proceedings of the 37th International Conference on Machine Learning (ICML'20), Vol. 119. JMLR.org, Article 276, 2943–2952.
>
> [6] Zhang, Y., Zhao, J., Liao, Z., Wu, W., Michieli, U., & Cannistraci, C. V. (2024). Brain-Inspired Sparse Training in MLP and Transformers with Network Science Modeling via Cannistraci-Hebb Soft Rule. Preprints. https://doi.org/10.20944/preprints202406.1136.v1
>
> [7] Zhang, Y.; Zhao, J.; Wu, W.; Muscoloni, A.; Cannistraci, C. V. Epitopological Learning and Cannistraci-Hebb Network Shape Intelligence Brain-Inspired Theory for Ultra-Sparse Advantage in Deep Learning. The Twelfth International Conference on Learning Representations (ICLR) 2024.

---

> ### Author Response · Authors · 2025-12-02
> **Reply to the Reviewer iDtv**
>
> **Reply to question 2**
>
> We sincerely thank the reviewer for raising this important point regarding baseline optimization. Our fully-connected (FC) baseline strictly follows the implementation described in **[1]**, with both its architecture and training pipeline adhering to well-established practices in the spiking neural network community. This ensures that the baseline represents a strong and reproducible reference point. For both the sparse and dense models, we maintained an identical experimental configuration across all aspects except for the introduced sparsity itself. This approach ensures a fair and direct comparison by eliminating any potential confounding variables.
>
> Building upon this rigorously validated baseline, we attribute the observed performance gain to CH-SNN's capability in mitigating overfitting. The CIFAR-100 dataset, with its 100 fine-grained categories and only 50,000 training images, is particularly susceptible to overfitting. By enforcing learning under a high sparsity regime—where 74.45% of connections are removed—CH-SNN compels the model to concentrate on the most discriminative and robust features. It is worth emphasizing that our approach does not merely introduce random sparsity. Instead, the dynamic pruning and CH3-L3 epitopological learning regeneration mechanism work collaboratively to continuously and dynamically refine the network architecture, eliminating redundant connections while regenerating functionally important ones. This process effectively uncovers a more efficient and generalizable topology than the original fully-connected structure.
>
> The experimental results provide strong support for this hypothesis. On the training set, the dense network achieved an accuracy of 78.53%, compared to 79.48% for the sparse CH-SNN. More notably, on the test set, the sparse model attained 75.22%, significantly outperforming the dense baseline at 72.06%. This clear improvement in generalization underscores that our method not only preserves but enhances model performance under structured sparsity.
>
> A finally note is that a recent article accepted in NeurIPs 2025 reports that CHTs algorithms are a “brain network science modelling of sparse neural networks that enables Transformers and LLMs to perform as fully connected” **[2]**. Therefore, our results are in line with the ones already achieved in literature in other AI tasks and contexts.
>
> **Reply to question 3**
>
> We thank the reviewer for raising this important question. The observed discrepancy indeed exists, and it stems from differences in baseline models across methods. We would like to provide a full clarification:
>
> The result of Grad R is cited from Paper **[1]**, and this gain is calculated based on its specific baseline. Since the code for Grad R is not publicly available, we were unable to fully reproduce its training pipeline and baseline model.
>
> To ensure a unified and fair comparison across all methods, we adopted our own thoroughly optimized standard fully-connected baseline (95.07% accuracy). Although the improvement from CH-SNN appears smaller (+0.38%), it was achieved on a stronger baseline. Notably, CH-SNN attains: (1) A higher absolute accuracy (95.45% vs. 91.95%) (2) A higher sparsity level (94.73% vs. 75.00%). These results demonstrate the effectiveness of our method under a more rigorous and comparable experimental setup. Regarding the comparison with other methods, we followed the principles below and would like to clarify as follows:
>
> SD-SNN: Since the code is available, we reimplemented this method and compared it with our CH-SNN and the fully-connected baseline under identical training conditions, including hyperparameters, and data preprocessing. This ensures a completely fair comparison with SD-SNN.
>
> Other methods (e.g., Grad R): As their code has not been released, we were unable to fully reimplement their training pipelines. Therefore, the experimental results for these methods, including both their baseline performance and sparse model performance, are directly adopted from their original papers.
>
> We confirm that both our baseline method and all sparse training approaches were evaluated under identical training configurations. These comparisons are therefore fair and reliable. To address the Reviewer concern, we now added a section in the **Appendix A.10** that summarizes and reports these guidelines to reproduce and compare the results.
>
> **References**
>
> [1] 	Bing Han, Feifei Zhao, Wenxuan Pan, Yi Zeng, Adaptive sparse structure development with pruning and regeneration for spiking neural networks, Information Sciences, Volume 689, 2025, 121481, ISSN 0020-0255, https://doi.org/10.1016/j.ins.2024.121481.
>
> [2] 	Zhang, Y., Cerretti, D., Zhao, J., Wu, W., Liao, Z., Michieli, U., & Cannistraci, C.V. (2025). Brain network science modelling of sparse neural networks enables Transformers and LLMs to perform as fully connected. Thirty-Ninth Conference on Neural Information Processing Systems (NeurIPS 2025).

---

### Author Response · Authors · 2025-12-03
**General Reply and Updated Manuscript**

**Summary**

We thank the reviewers for their valuable time and feedback, which helps us to  strengthen our paper. Overall, the reviewers highlighted several key strengths of our submission: (1) The novelty of integrating Cannistraci-Hebb theory from network science into SNN sparse training (9DG8, nSp9). (2) The introduction of specialized initialization schemes (SSCTI, SSWI) tailored for spike-based learning (nSp9). (3) Comprehensive experimental validation across multiple datasets and architectures, including thorough ablations, sensitivity analyses, and hardware efficiency evaluations (zWas, 9DG8, nSp9). (4) A clear, modular four-stage framework that is intuitive and extensible (9DG8) and robust performance in achieving ultra sparsity ($\geq99\\%$) without degradation, supported by well-structured experiments (iDtv, zWas).

To address the reviewers' concerns, we have made the following key improvements during the rebuttal period.

**Additional Experiments**
- In response to the concerns about evaluation on complex datasets and deeper networks (iDtv, 9DG8), we have conducted extensive supplementary experiments on Tiny-ImageNet and ImageNet using ResNet-18-based SNNs. The experimental results demonstrate CH-SNN's competitive accuracy (e.g., only 0.85% drop on ImageNet at 66.94% sparsity) and strong scalability. We have added a section in the **Appendix A.12** that summarizes and reports experimental results.
- We have coducted new experiments on varying sparsity ratios (70-99%), pruning frequencies (dynamic/static), regrowth sampling ($\delta=0/0.5/1$), timesteps ($T=2-16$), and robustness under noise types (bit-flip, false-spike, dropout), quantifying latency reductions (up to two orders of magnitude) and Relative Performance Degradation Rate (RPDR), to show CH-SNN's robustness advantages on native spiking data (zWas, 9DG8, nSp9). More details are shown in **Appendix A.7**, **Appendix A.9** and **Appendix A.11**.

**Clarifications**
- We have clarified that methods with available code (e.g., SD-SNN) were reimplemented under identical conditions, while others (e.g., Grad R) used reported results from original papers. We have added a section in the **Appendix A.10** that summarizes and reports accurately these guidelines to reproduce and compare the results.
- We have added the specified Spikformer architecture details (8 layers, 512 hidden dim, etc.) to the **Appendix A.5**.
- We have added formal mathematical formulations in **Appendix A.3** for pruning and regrowth processes, including equations for probabilities, multinomial sampling, and regrowth scores.
- We have added a section to **Appendix A.2.1** that summarises and accurately reports how the CH3-L3 regrowth mechanism guarantees stability and avoids redundant regrowth loops.
- We have added **Appendix A.9** for robustness analysis and **Appendix A.4** detailing the SSCTI workflow.
- We have added a section in the **Appendix A.2.2** to discuss the biological interpretability of CH3-L3 topological regrowth.

These changes are incorporated into the revised manuscript, with new appendices and tables for clarity. We believe they fully address the reviewers' points and elevate the paper's quality. Thank you for considering our submission.

---

### Meta-Review · Area_Chair_vvS6 · 2025-12-21

**Summary:**

This paper proposes a four-phase dynamic sparse training algorithm for spiking neural network (SNNs) inspired by Cannistraci-Hebb theory of complex network science. It integrates with the sparse topology initialization, spike-aware weight initializations, hybrid pruning and topological regrowth in order to achieve good structure sparsities that improve accuracies. Decoding is based on event-based and frame-based vision tasks, architecture (CNNs and Spikformer) and hardware efficiency analysis allows spanning over different datasets.

The reviewers appreciated the novelty, structure, comprehensive experimental studies and particularly the interdisciplinary fusion between network science and neuromorphic computing. First some of the initial concerns were on scalability to larger dataset, comparison fairness (is not just pruning/regrowth math transparency), regrowth stability and gain interpretation on saturated benchmarks. The authors provided a strong rebuttal including additional experiments at larger scale (Tiny-ImageNet and ImageNetd), formalizing the mechanisms, performing sensitive analysis and clarifying protocols.

**Reviewer Concerns:**

Shared comments:

1. Scaling to larger Datasets and Networks. Reviewers were skeptical whether ultra-sparsity is preserved outside of toy-benchmarks (e.g., MNIST) and shallow models, or if it breaks down on more orders-of-magnitude data. Resolution: The authors included Tiny-ImageNet and ImageNet experiments with SNNs based on ResNet-18, compared to DPAP. Experiments show that it induces sustained sparsity (67%) while achieving only marginal accuracy degradation (0.85% on ImageNet).

2. Fairness and Transparency of Comparisons. Reviewers asked for information on how good the reimplementations were relative to baselines in terms of reported numbers and whether gains are due to non-reproducible setups. Resolution: the authors provided reimplemented methods; an appendix on protocols was added and non-reproducible cases were justified, such that reproducibility and fairness are increased.

3. Performance Gains Under Extreme Sparsity. There was skepticism from the reviewers over modest gain on fully saturated datasets (such as MNIST), may be misleading. Resolution: The authors indicated there was a focus on performance at extreme sparsity (~98%) where collapse is prevalent. They pointed out better performance on CIFAR-10/100 and event-based datap, shifting the emphasis accordingly.

4. Stability, convergence and mathematical clarity of Pruning/Regrowth. Reviewers asked for more precise math on pruning/regrowth, antioscillation guarantees and stability proof. Resolution: Formulae, sampling details (multinomial), early- stop criteria, theoretical intuitions and empirical stability proofs have been provided by the authors. Our sensitivity analysis on sparsity ratios, pruning frequency and regrowth distribution further supports our findings are robust.

Reviewer-Specific Questions:

1. Reviewer iDtv: Highlighted scalability, Spikformer clarity, and marginal gain. Authors added large-scale experiments (Tiny-ImageNet/ImageNet), Spikformer details, and explained why gains under sparsity are significants.

2. Reviewer zWas: Sought pruning/regrowth math and sparsity/threshold sensitivity. Authors added formulations, sampling explanations, and extensive experiments on temporal/structural sparsity and spike threshold.

3. Reviewer 9DG8: Questioned CH3-L3 regrowth stability, hyperparameter sensitivity, timestep choice, and biological grounding. Authors offered theoretical/empirical stability arguments, hyperparameter sweeps, timestep ablations, and nuanced biological discussion based on neuroscientific literatured.

4.Reviewer nSp9: Inquired about temporal sparsity-latency trade-offs and reproducibility. Authors added latency/robustness analyses under varying sparsiti and clarified baselines reproduction.

**Reviewer Scores:**

I think the scores from all reviewers are reasonable.

---

### Decision · Program_Chairs · 2026-01-26

Accept (Poster)